# Oligodendrocyte dynamics dictate cognitive performance outcomes of working memory training in mice

Takahiro Shimizu [1,7], Stuart G. Nayar [1,7], Matthew Swire [1], Yi Jiang [1], Matthew Grist [1], Malte Kaller [2], Cassandra Sampaio Baptista [2,6], David M. Bannerman [3], Heidi Johansen-Berg [2], Katsutoshi Ogasawara [4], Koujiro Tohyama [5], Huiliang Li [1] & William D. Richardson [1] ✉

Previous work has shown that motor skill learning stimulates and requires generation of myelinating oligodendrocytes (OLs) from their precursor cells (OLPs) in the brains of adult mice. In the present study we ask whether OL production is also required for non-motor learning and cognition, using T-maze and radial-arm-maze tasks that tax spatial working memory. We find that maze training stimulates OLP proliferation and OL production in the medial prefrontal cortex (mPFC), anterior corpus callosum (genu), dorsal thalamus and hippocampal formation of adult male mice; myelin sheath formation is also stimulated in the genu. Genetic blockade of OL differentiation and neo-myelination in *Myrf* conditional-knockout mice strongly impairs training-induced improvements in maze performance. We find a strong positive correlation between the performance of individual wild type mice and the scale of OLP proliferation and OL generation during training, but not with the number or intensity of c-Fos+ neurons in their mPFC, underscoring the important role played by OL lineage cells in cognitive processing.

Oligodendrocytes (OLs), the myelin-forming cells of the central nervous system (CNS), are generated during development from OL precursors (OLPs), which arise in localized parts of the ventricular zones of the embryonic brain and spinal cord before proliferating and migrating widely to become almost uniformly distributed through the postnatal and adult CNS[1]. Most myelinating OLs are formed in the early postnatal period (first ~6 postnatal weeks in mice) but OLPs continue to divide and generate new myelinating OLs throughout adulthood.

During adulthood, OLPs and newly forming OLs can detect and respond to electrical activity in the axons that they contact. For example, OLPs express AMPA receptors, form physical synapses with axons and respond to glutamate released from active axons[2,3]. Neuronal activity stimulates OLP differentiation into OLs and/or survival of the newly forming OLs, thereby enhancing myelination of electrically active axons in preference to their inactive or less-active neighbours[4,5]. Electrical activity or experience can also influence the number of myelin sheaths synthesized by individual OLs, or myelin sheath length or thickness[5–9]. These different sorts of modification alter the properties of neural circuits in response to physiological demand and are known collectively as "adaptive myelination".

Adaptive myelination has been shown to be important for learning and memory. McKenzie et al. [10] blocked the formation of

[1]Wolfson Institute for Biomedical Research, University College London, Gower Street, London WC1E 6BT, UK. [2]Wellcome Centre for Integrative Neuroimaging, Department of Clinical Neurosciences, John Radcliffe Hospital, University of Oxford, Oxford OX3 9DU, UK. [3]Department of Experimental Psychology, University of Oxford, Oxford OX1 3TA, UK. [4]Technical Support Center for Life Science Research, Iwate Medical University, 1-1-1 Idaidori, Yahabacho, Shiwa-gun, Morioka, Iwate 028-3694, Japan. [5]Department of Physiology, Iwate Medical University, 1-1-1 Idaidori, Yahabacho, Shiwa-gun, Morioka, Iwate 028-3694, Japan. [6]Present address: Institute of Neuroscience and Psychology, University of Glasgow, 62 Hillhead Street, G12 8QB Glasgow, UK. [7]These authors contributed equally: Takahiro Shimizu, Stuart G. Nayar. ✉e-mail: w.richardson@ucl.ac.uk

newly-forming OLs by conditional knockout (cKO) in OLPs (using *Pdgfra-CreER^T2*) of *Myelin regulatory factor* (*Myrf*), encoding a transcription factor that is necessary for OL differentiation. The *Myrf*-cKO mice were impaired at learning a new motor skill – running at speed on a "complex wheel" with unevenly spaced rungs. When wild-type mice learned to run on the complex wheel, OL differentiation and/or survival was stimulated as early as a few hours into learning[10,11]. Pan et al.[12] and Steadman et al.[13] used a similar *Myrf*-cKO approach (using *NG2-CreER^TM*) in contextual fear-conditioning paradigms and found that new OL production was required for the formation and recall of long-term (28 day) fear memory, though not for fear learning per se or for short-term recall. In addition, Steadman et al.[13] found that long-term spatial memory in the Morris water maze was impaired although, again, spatial learning per se was relatively unaffected. Together, these studies suggest that new myelin is required to modify and stabilize task-relevant circuits, with short and longer-term behavioural consequences that might be context-dependent.

Motor skill learning engages motor cortex, basal ganglia, cerebellum and other brain regions but is independent of the hippocampus[14]. On the other hand, fear learning and spatial learning require cognitive function and rely on coordinated activity of the hippocampus and other brain regions, but do not rely on OL generation. We were therefore driven to ask whether there are examples of non-motor learning that do depend on OL genesis. We chose to investigate the role of OL genesis in cognition, in particular the improvement in cognitive performance that can accompany working memory training in mice. This choice was influenced by the fact that working memory training in humans induces microstructural changes in white matter tracts[15], similar to motor skills learning[16,17] and consistent with a role for adaptive myelination.

Working memory is a short-term, limited-capacity memory system that in humans is crucial for cognitive processes involved in decision-making and reasoning[18–21]. Working memory engages the frontoparietal network, together with its long-range interconnections through the corpus callosum and other tracts. Spatial working memory tasks additionally engage hippocampal circuits including the fimbria-fornix[22,23]. Working memory capacity/ performance in a given task can be improved (trained) through reiterative practice both in humans[24–27] and mice[28], although how training modifies the underlying psychological processes and neural circuits is not known.

We investigated the role of OL genesis in the performance of mice in T-maze and 8-arm radial maze tests, using "win-shift" protocols that train and assess spatial working memory[29–33]. We found that *Myrf*-cKO mice were unable to improve their performance in either a delayed non-matching to position (DNMP) T-maze task (rewarded alternation) or an analogous radial arm maze task, relative to control littermates, which improved their performance steadily over the 8- or 9-day training period. During maze training, wild-type mice increased OLP proliferation and production of newly differentiated OLs in the prefrontal cortex and hippocampal formation, especially their long-range connecting axon tracts in the anterior corpus callosum (genu) and fimbria. By immunofluorescence light microscopy and electron microscopy we also obtained evidence that the number of myelin sheaths and associated nodes of Ranvier in the genu were increased during training. Hence, working memory training, like motor skills training, both stimulates and requires active OL generation and myelin formation.

It was striking that the performance of individual animals in the radial arm maze correlated closely with the rate of OLP division and the number of newly generated OLs that appeared in the genu and mPFC during training. These changes in OL lineage dynamics were not mirrored by population-level changes in neuronal activity, estimated either by the number or the average fluorescence intensity of recently active cFos-expressing neurons in the mPFC. Our findings indicate that OL generation and neo-myelination strongly influence cognitive ability, by strengthening structural connectivity or coordinating activity within and among distributed brain regions involved in working memory operations.

## Results

Adult oligodendrocyte generation is stimulated by, and required for, various learning and memory tasks including motor learning, long-term spatial memory and remote fear memory[6,10–13,34]. We asked whether oligodendrocyte generation is also required for learning paradigms that rely on working memory performance.

### Active OL generation is required for working memory training in the T-maze

The T-maze rewarded alternation task (Methods and Supplementary Video 1) uses a delayed non-matching to position (DNMP) protocol to assess spatial working memory and training-induced improvement in working memory performance over the duration of the task[35]. *Myrf*-cKO mice (*n* = 28) and littermate controls (*n* = 26) were placed on dietary restriction one week prior to the 3 days of habituation and 8 days of training/ testing in the T-maze (Methods and Fig. 1A). On day 1 of training, *Myrf*-cKO mice, which cannot generate new OLs post-tamoxifen administration[10], performed at near-chance levels (~50% success rate) and did not improve significantly during the 8 days of training (Fig. 1B). In contrast, control littermates started at chance levels but steadily improved over the next 8 days, reaching significant divergence from *Myrf*-cKOs on days 7 and 8 (Fig. 1B). These results suggest that active generation of new OLs is required for training-based improvement (learning) in the rewarded alternation task.

### OL generation is not required for simple left-right discrimination in the T-maze

Mice were also trained in a simple, appetitively-motivated left-right discrimination task using the same T-maze apparatus. This task has the same sensorimotor and motivational demands as the spatial working memory task described above. The same goal arm was baited with a food reward (dilute condensed milk) on each trial during the full 3 days of the experiment for each mouse. Mice were released into the start-arm and had to choose whether to turn left or right at the T-junction in order to obtain the reward. Those that chose the nonbaited arm were recorded as having made a reference memory error. On day 4 the location of the rewarded goal-arm was switched so that the other arm was now always baited, requiring the mice to adapt and turn in the opposite direction than before. Left and right goal-arms were counterbalanced among mice of both genotypes. Both groups of mice successfully learned the task and then subsequently learned to reverse their choice of arm. The performances of the control and *Myrf*-cKO groups were superimposable, both before and after the reversal of goal arms (Supplementary Fig. S1A). Therefore, active OL generation is not required for left-right discrimination learning. This suggests that the spatial working memory deficit reported above was not due to impaired sensorimotor or motivational processes, or inability to discriminate between the arms of the T-maze.

### OL generation is not required for recognition memory

Mice were then assessed on two spontaneous, exploratory tasks of recognition memory - the novel object recognition task (NOR) and the object location task (OLT). There was no difference in the performance of *Myrf*-cKOs versus control littermates in the NOR, whether tested 10 min or 24 h after first encountering the objects (Supplementary Fig. S1B, C). There also was no difference between *Myrf*-cKOs and controls in the OLT after 10 min (Supplementary Fig. S1D, E). Thus, active OL generation is not required for spatial or object recognition memory. Taken together, these data indicate that *Myrf*-cKO mice can learn to recognize either objects or locations as familiar.

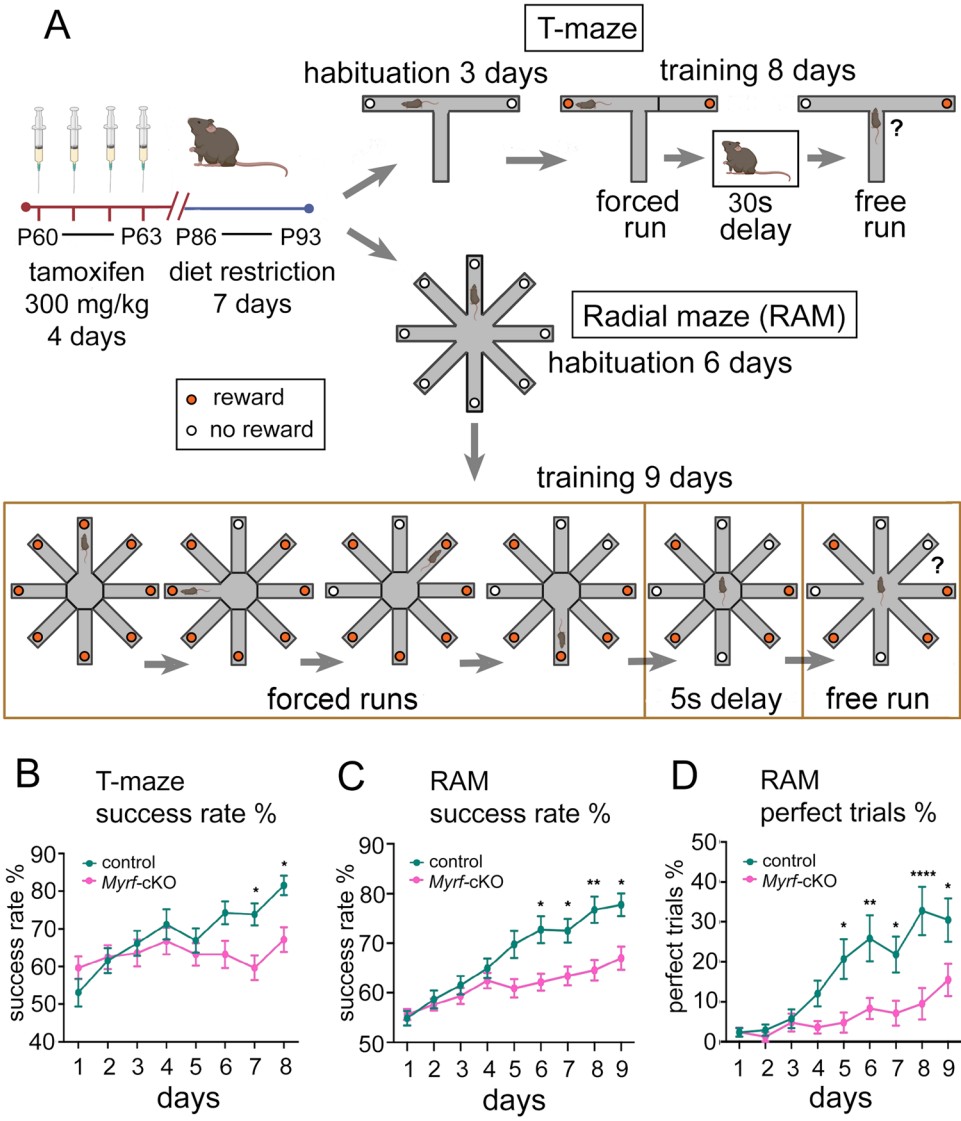

**Fig. 1 | Working memory training requires OL generation. A** T-maze and radial arm maze (RAM) protocols (drawing created using BioRender). **B** Success rates of control ($n = 26$) and *Myrf*-cKO ($n = 28$) adult male mice during T-maze training. Controls improved their success rate over the 8 days of training whereas *Myrf*-cKOs barely improved [repeated measures 2-way ANOVA: time x genotype $p = 0.0012$, $F(7, 364) = 3.50$; time, $p < 0.0001$, $F(7, 364) = 7.16$; genotype, $p = 0.066$, $F(1, 52) = 3.52$]. Controls attained a significantly greater success rate on the final two days of T-maze training compared to *Myrf*-cKOs (Day 7: control 74% ± 2.9%, *Myrf*-cKO 60% ± 3.3%, $p = 0.02$. Day 8: control 82% ± 2.6%, *Myrf*-cKO 67% ± 3.3%, $p = 0.009$, Šídák's post-test). **C** Success rates of control ($n = 28$) and *Myrf*-cKO ($n = 29$) adult male mice over 9 days of RAM training [repeated measures 2-way ANOVA: time x genotype $v < 0.0001$, $F(8, 440) = 7.41$; time $p < 0.0001$, $F(4,$ 239) = 39.8; genotype $p = 0.009$, $F(1, 55) = 7.5$]. Controls surpassed *Myrf*-cKOs on days 6-9 (e.g. Day 8: control 77% ± 3%, *Myrf*-cKO 65% ± 2% $p = 0.006$. Day 9: control 78% ± 2%, *Myrf*-cKO 67% ± 2%, $p = 0.02$. Šídák's post-test). **D** Fraction (%) of trials over the full 9 days of RAM testing in which mice recorded no working memory errors ("perfect trials") [repeated measures 2-way ANOVA: time x genotype $p < 0.0001$, $F(8, 440) = 5.4$; time $p < 0.0001$, $F(8, 440) = 19$; genotype $p = 0.006$, $F(1, 55) = 8.1$]. Control mice ($n = 29$) recorded more perfect trials than *Myrf*-cKOs ($n = 28$) on days 5-9 (e.g. Day 8: control 33% ± 6%, *Myrf*-cKO 10% ± 4%, $p < 0.0001$. Day 9: control 31% ± 5%, *Myrf*-cKO 16% ± 4%, $p = 0.03$, Šídák's post-test.). Data are presented as mean ± s.e.m. n.s. not significant ($p > 0.05$), *$p \leq 0.05$, **$p \leq 0.01$, ****$p \leq 0.0001$. Source data are provided as a Source Data file.

In addition, a Y-maze was used to assess short-term spatial recognition memory (Methods). *Myrf*-cKO and control mice were allowed to explore and become familiar with 2 arms of the Y-maze (the "start" and "other" arms), access to the remaining ("novel") arm being blocked. They were then removed from the maze to their home cage for 2 min, then returned to the maze and allowed to explore all 3 arms at will. The number of arm entries and time spent in each arm were recorded and a "discrimination ratio" [novel arm/ (novel + other arm)] calculated. By this measure *Myrf*-cKO mice displayed similar spatial novelty preference as the controls (Supplementary Fig. S1F). This suggests that mouse orientation based on spatial location and/or distal spatial cues is unimpaired by *Myrf* deletion. In addition, since mice must remember which arms they visited previously in order to reveal their preference for the unvisited "novel" arm, this experiment demonstrates that short-term spatial recognition memory is unimpaired in *Myrf*-cKOs, so their failure to improve performance with training in the T-maze rewarded alternation task is not down to a general deficit in baseline short-term memory.

We also tested *Myrf*-cKO mice in the open field test for 10 min (Methods). There were no significant differences in mean speeds, distances travelled, or trajectories in bird nest and heat maps (Supplementary Fig. S1G-I). Therefore, *Myrf*-cKO mice are not hypo- or hyper-active compared to controls.

## OL generation is required for working memory training in the 8-arm radial maze

To test the generality of the spatial working memory deficit that we observed on the T-maze, in a more complex spatial environment with increased memory demands, we also assessed experimentally naïve mice on an 8-arm radial maze (RAM) task, which requires mice to visit 4 assigned arms sequentially during forced runs before then selecting the 4 unvisited arms from all 8 possible options during the free run phase of the task.

Our RAM protocol consisted of 6 days habituation followed by 9 days of training/ testing (Methods and Fig. 1A). During days 1-4 of training the control mice ($n = 29$) and *Myrf*-cKOs ($n = 28$) both improved their performance at a similar rate and reached a similar level of performance, assessed either by success rate or percent of "perfect trials" (Methods and Fig. 1C, D). The control group out-performed the *Myrf*-cKOs by success rate on each of days 5 to 9 (e.g. day 8: controls 76.7% ± 2.6%, *Myrf*-cKOs 64.6% ± 2.1%, $p = 0.006$, Šídák's post-test) (Fig. 1C). The proportion of "perfect trials" achieved by controls also exceeded that of the *Myrf*-cKOs on each of days 5 to 9 (e.g. day 8: controls 32.8% ± 6.1%, *Myrf*-cKOs 9.5% ± 4.0%, $p = 0.0001$) (Fig. 1D). The proportion of perfect scores over all 9 testing days was also significantly higher in the control group (9.4% ± 1.8% versus 3.5% ± 1.0%, $p = 0.004$). The distance travelled by *Myrf*-cKO mice over the 9 days of RAM training and their average running speed were the same as controls (Supplementary Fig. SJ, K).

There are at least two strategies that mice can adopt during the free run phase of the RAM task: 1) they can use working memory to identify unvisited arms and collect the remaining rewards directly, or 2) they can visit all arms sequentially, clockwise or anticlockwise, until they collect all the rewards ("daisy-chaining"). This latter approach, which does not tax working memory, was used by the majority of mice, both *Myrf*-cKOs and controls, during the first few days of the task (Supplementary Videos 2-4). A proportion of the control mice (not the *Myrf*-cKOs) subsequently switched to the more efficient working memory-based approach (Supplementary Video 2). By manual modelling we determined that the average score that can be achieved over many trials by daisy-chaining is independent of whether mice run clockwise or anti-clockwise and ranges between 53% and 56% (mean, 55%) for the 4-arm forced run patterns that we employed (see Methods). This is close to the starting score we observed on day 1 for both *Myrf*-cKOs and controls, consistent with both groups initially using the daisy-chain strategy (Fig. 1C). However, the final average scores attained on days 8 and 9 by the control group (77% and 78% respectively) cannot be achieved by daisy-chaining and must rely on working memory. This is also suggested by the rapidity of correct goal-arm selection in the latter stages of training (Supplementary Video 2). Note that daisy-chaining can never result in a "perfect trial" with the 4-arm patterns that we employed. Therefore, our RAM data (Fig. 1C, D) strongly imply that *Myrf*-cKO mice cannot train their working memory to the same degree as normal controls and that de novo OL generation is a critical factor in training-induced working memory improvement.

There was no evidence that the spatial working memory deficit in *Myrf*-cKO mice was due to an increased susceptibility to proactive interference. We re-plotted the data of Fig. 1C, separating the first 3 trials from the last 3 trials of each day (Supplementary Fig. S1L). The choice accuracy of control mice was not different between the earlier or later trials of each day, nor was the choice accuracy of *Myrf*-cKOs. This suggests that *Myrf*-cKO mice were no more likely than their control littermates to confuse arm visits made in their current trial with visits made in previous trials, so this was not a factor in the under-performance of *Myrf*-cKOs in the RAM.

## Working memory training stimulates OLP division and new OL generation in the anterior corpus callosum

Motor skill learning is known to stimulate OLP proliferation and OL generation so we asked whether the same is true of working memory training. To label and visualize newly generated OLs we administered EdU to phenotypically wild-type mice via their drinking water (0.2 mg/ml) during 9 days of training in the radial arm maze task. A control group remained in their home cages throughout (home-cage controls). OL lineage cells were analyzed either 1- or 14-days post-training, by immunolabelling for Pdgfra (to visualize OLPs) or with monoclonal CC1 (to visualize differentiated OLs) together with EdU labelling to detect recently divided, newly-generated cells (Fig. 2).

Working memory engages the medial prefrontal cortex (mPFC), including the anterior cingulate cortex (ACC) and prelimbic/ infralimbic cortex (PLC/ ILC), together with their inter-hemispheric connections in the anterior-most corpus callosum (anterior CC, also known as the genu) and relay centres in the mediodorsal thalamus (MDT). Spatial working memory also involves the hippocampal formation including CA1 and the fimbria (Fim). Therefore, we analyzed OLP proliferation and differentiation into OLs in these different brain regions of "good-performing", "poor-performing" and home-cage control mice. Good-performers were mice that achieved ≥10 perfect trials during the 9 days of training, poor-performers ≤5 perfect trials. Poor-performers experienced the same handling and exposure to the RAM as good-performers but performed less well; they therefore provided an ideal control group for separating genuine learning effects of RAM-training from potentially confounding effects of differing experience and activity. Data for genu and ACC are shown in Fig. 3 and Supplementary Data 1); data for PLC/ ILC, MDT, hippocampal CA1 and fimbria (Fim) are given in Supplementary Fig. S2 and Supplementary Data 2).

The proliferation of OLPs was dramatically increased ( ~ 4-fold) in good-performers compared to either poor-performers or home-cage controls in the anterior corpus callosum, judging by the number-density of EdU$^+$ Pdgfra$^+$ cells at 1-day post-RAM training (Fig. 3B). Consequently, the population density of Pdgfra$^+$ OLPs was also significantly increased in good-performers at 1-day post-RAM (Fig. 3C). The number-density of EdU$^+$, CC1$^+$ newly-differentiated OLs was also significantly increased at 1-day post-RAM in good-performers compared to poor-performers or home-cage controls (Fig. 3D; statistics in Supplementary Data 1). Because of when EdU was administered, stimulation of OLP proliferation must have occurred during the 9 days of RAM working memory training.

By 14-days post-RAM training, the number-density of EdU$^+$ Pdgfra$^+$ OLPs in good-performers had dropped to a similar level as in poor-performers or home cage controls (Fig. 3E). This likely reflects differentiation of a fraction of recently-divided OLPs into EdU$^+$ CC1$^+$ OLs, because the number of EdU$^+$ OLs at 14-days post-RAM is similar to the number of EdU$^+$ OLPs at 1-day post-RAM, while the number of EdU$^+$ OLPs drops to near baseline (compare Fig. 3B, E, G). Unexpectedly, the overall population density of Pdgfra$^+$ OLPs in good-performers remained elevated at 14-days post-RAM (Fig. 3C, F) despite the number of newly-divided EdU$^+$ Pdgfra$^+$ OLPs having fallen to baseline. Presumably, Pdgfra$^+$ OLPs continued to divide and accumulate after the RAM training period, when EdU was no longer available. Whether, over the longer term, all these excess EdU-negative OLPs eventually differentiate into OLs and return the OLP population density to pre-training levels is an intriguing question.

We observed similar effects of RAM-training on OL lineage dynamics in the fimbria, a white matter tract that connects the hippocampus to its major output regions and is required for spatial aspects of learning[36] including spatial working memory (but not spatial reference memory)[22]. Here too, RAM-training stimulated OLP proliferation leading to an increased number of EdU$^+$ Pdgfra$^+$ OLPs at 1-day post-RAM in good-performers relative to controls, and an elevated number of newly-formed EdU$^+$ CC1$^+$ OLs that persisted until at least 14-days post-RAM (Supplementary Fig. S2; statistics in Supplementary Data 2).

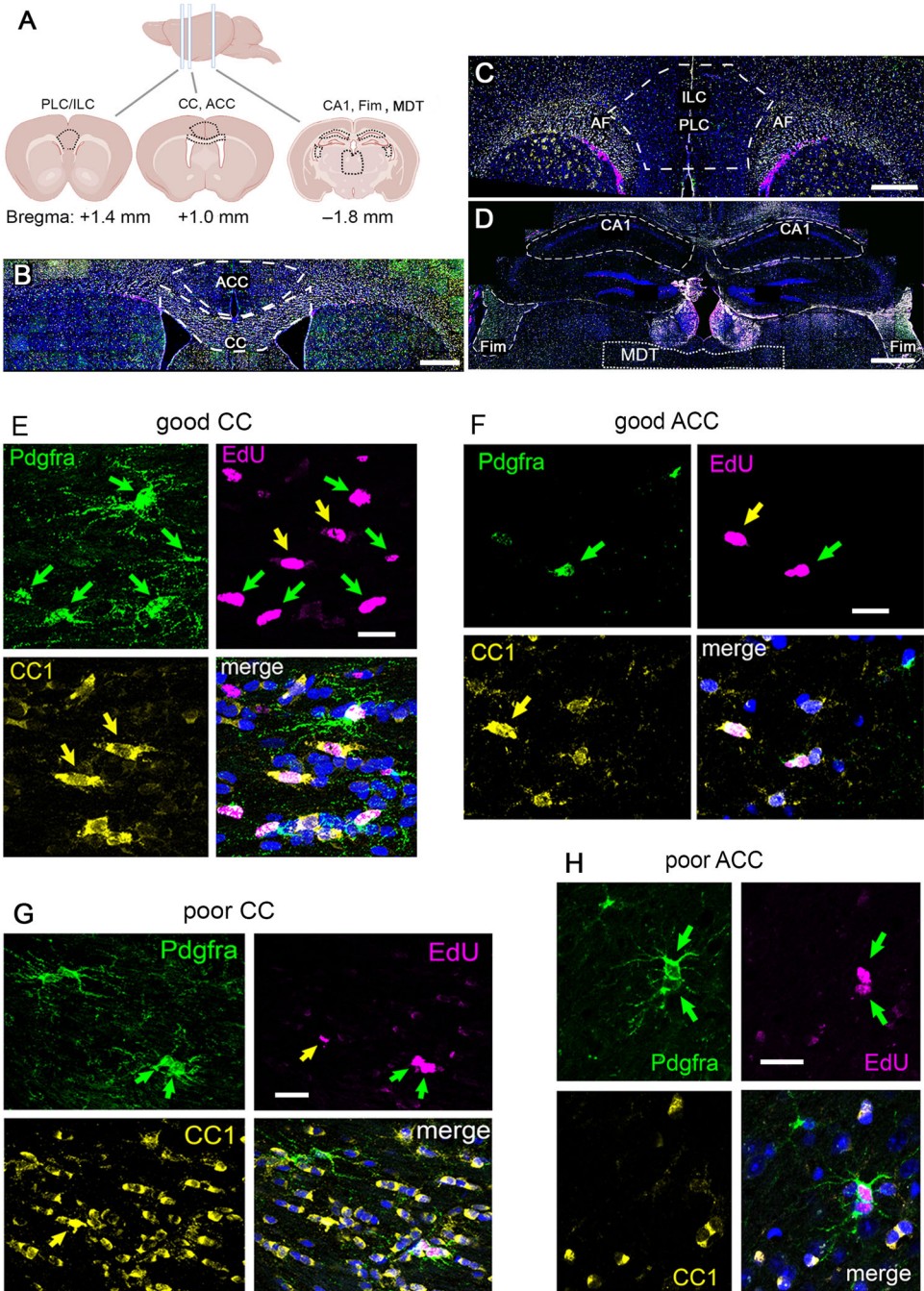

**Fig. 2 | Immunofluorescence analysis of OL lineage cells. A** Coronal sections through the brains of RAM-trained mice (1- or 14-days post-training) or home cage controls were analyzed at the level of the anterior cingulate cortex (ACC) ( - Bregma +1.0 mm), prelimbic/ infralimbic cortex (PLC/ILC) ( - Bregma +1.4 mm), hippocampus (CA1), fimbria (Fim) and mediodorsal thalamus (MDT) ( - Bregma –1.8 mm) (Schematic created using BioRender). OL lineage cells were identified by immunolabelling with anti-Pdgfra (for OLPs) and monoclonal CC1 (for differentiated OLs), together with EdU histochemistry to identify recently-divided cells and their progeny. Sections were post-stained with Hoechst dye (blue) to label cell nuclei. **B**-**D** Low-magnification images illustrating the areas analyzed: (**B**) ACC and underlying anterior corpus callosum (CC), (**C**) PLC/ILC, (**D**) hippocampal CA1, Fim and MDT. **E**-**H** Representative higher-magnification images of OL lineage cells in the ACC and CC of mice that we categorized as either good- (**E**, **F**) or poor-performers (**G**, **H**) in the RAM task (≥10 or ≤5 "perfect trials" during the 9 days of RAM training/testing; *n* = 6 good- and *n* = 6 poor-performers). *Green arrows*, recently-divided EdU⁺ Pdgfra⁺ OLPs; *yellow arrows* newly-formed EdU⁺ CC1⁺ OLs. Micrographs are representative of more than 3 independent immunolabelling experiments. *Scale bars:* (**B**–**D**), 1 mm; (**E**–**H**), 20 µm.

## RAM training stimulates OLP proliferation and OL genesis in the prefrontal cortex

In control mice, the steady-state population density of Pdgfra⁺ OLPs in the gray matter of the ACC is around half of that in the underlying white matter (Fig. 3C, F, I, L) while the rate of OLP proliferation and EdU incorporation is around ten-fold less in gray than in white matter (Fig. 3B, E, H, K). Nevertheless, at 1-day post-training, EdU incorporation into Pdgfra⁺ OLPs in the ACC was strongly increased ( ~ 2- to 7-fold) in good-performers relative to poor-performers or home cage controls (Fig. 3H). This proliferative response also results in an increase in the population density of Pdgfra⁺ OLPs in good-performers (Fig. 3I) and an increase in production of newly-differentiated EdU⁺ CC1⁺ OLs (Fig. 3J) at 1-day post-RAM. However, these increases are temporary and short-lived in the ACC; by 14-days

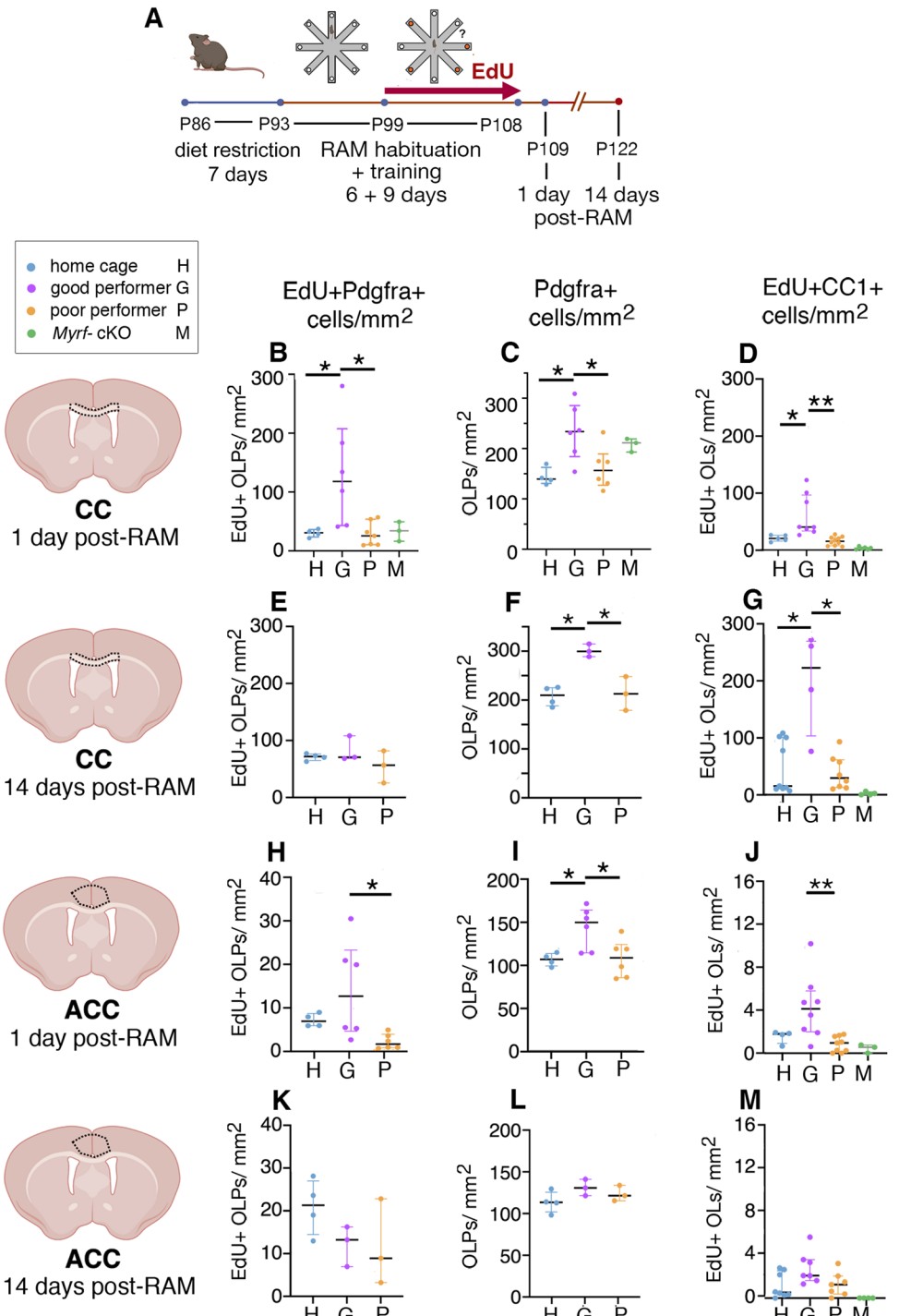

post-RAM there were no longer any detectable differences among groups, except for a non-significant trend towards an increase in newly-formed CC1+ OLs in good-performers (Fig. 3K-M; statistics in Supplementary Data 1).

Similar short-term effects of RAM-training on OL population dynamics were observed in another part of the medial PFC, the PLC/ILC (Supplementary Fig. S2B-G). In hippocampal CA1 there might have been a transient training-induced stimulation of OLP proliferation at 1-day post-RAM but no evidence of increased OL differentiation (Supplementary Fig. S2H-M); numbers of EdU+ CC1+ OLs were around 10-fold less than the number of EdU+ Pdgfra+ OLPs both at 1- and 14-days post-RAM, so most of the daughters of OLP division must have

failed to differentiate, or else differentiated and died. OL dynamics in the MDT (Supplementary Fig. S2T-Y) resembled those in CA1.

At 1-day post-training, there were similar numbers of EdU+ Pdgfra+ OLPs in Myrf-cKO mice as in poor-performers or home cage controls (Fig. 3B) and there were no significant differences in total numbers of Pdgfra+ OLPs among Myrf-cKOs, poor-performers and home cage controls (Fig. 3C). Almost no EdU+ CC1+ OLs were found in either the corpus callosum or pre-frontal cortex (ACC and PLC/ILC) of Myrf-cKOs that had undergone RAM training (Fig. 3D, G, J, M; Supplementary Fig. S2D, G), as expected[10,12,13]. The poor performance of Myrf-cKOs in the RAM task (relative to good-performers) could have resulted either from the inherently low level of OLP proliferation in Myrf-cKOs (similar

**Fig. 3 | Successful working memory training stimulates proliferation and differentiation of OLPs. A** Experimental protocol. Mice were from our *Pdgfra-CreER[T2]: Myrf [(flox)]* breeding colony. *Myrf [(flox/flox)]* and some *Myrf [(flox/+)]* received tamoxifen as in Fig. 1A while other *Myrf [(flox/+)]* did not. They were given EdU in their drinking water during radial arm maze (RAM) training and perfusion-fixed 1- or 14-days post-training. RAM-trained mice were characterized as good- or poor-performers based on whether they achieved ≥10 or ≤5 "perfect trials", respectively, over the 9 days of RAM training. Home cage controls did not experience dietary restriction and were not exposed to the RAM at any time. **B-D** In the corpus callosum (CC) at 1-day post-RAM, the number-densities of proliferating OLPs (EdU⁺ Pdgfra⁺), all OLPs (Pdgfra⁺) and newly-formed OLs (EdU⁺CC1⁺) were all increased in good-performers relative to poor-performers. Note that in the best of the good-performers >90% OLPs proliferated (compare **B, C**). Poor-performers were indistinguishable from home cage controls. **E-G** In the CC at 14-days post-RAM densities of OLPs and newly-formed OLs were still elevated in good- versus poor-performers. The number of newly differentiated OLs was increased further from 1-day post-RAM because of

continuing OLP differentiation post-RAM (compare **D,G**). **H-J** Also in the anterior cingulate cortex (ACC) at 1-day post-RAM proliferating OLPs (EdU⁺Pdgfra⁺), all OLPs (Pdgfra⁺) and newly-formed OLs (EdU⁺CC1⁺) were all more numerous in good-versus poor-performers, but by 14-days post-RAM all had returned to baseline (**K-M**). Corresponding data for prelimbic/ infralimbic cortex, mediodorsal thalamus, hippocampal CA1 and fimbria are shown in Supplementary Fig. S2. Data for *Myrf*-cKO mice (**D, G, J, M**), were included here primarily as a technical control for the experiments in Fig. 1, so they are not included in the statistical analysis. As expected, almost no new EdU⁺CC1⁺ OLs were produced in the *Myrf*-KOs. (**B-M**) x-axis labels are: H=home cage control, G=good performer, P=poor performer, M=*Myrf*-cKO, as also indicated in the key beneath panel (**A**). Data are presented as median ± 25%-75% interquartile range. p-values were determined by the Kruskal-Wallis non-parametric test, corrected for multiple comparisons using the Benjamini-Krieger-Yekutieli (BKY) false discovery rate test[88]. *$p \le 0.05$, **$p \le 0.01$ (see Supplementary Data 1 for full statistics). Source data are provided as a Source Data file. Drawings were created using BioRender.

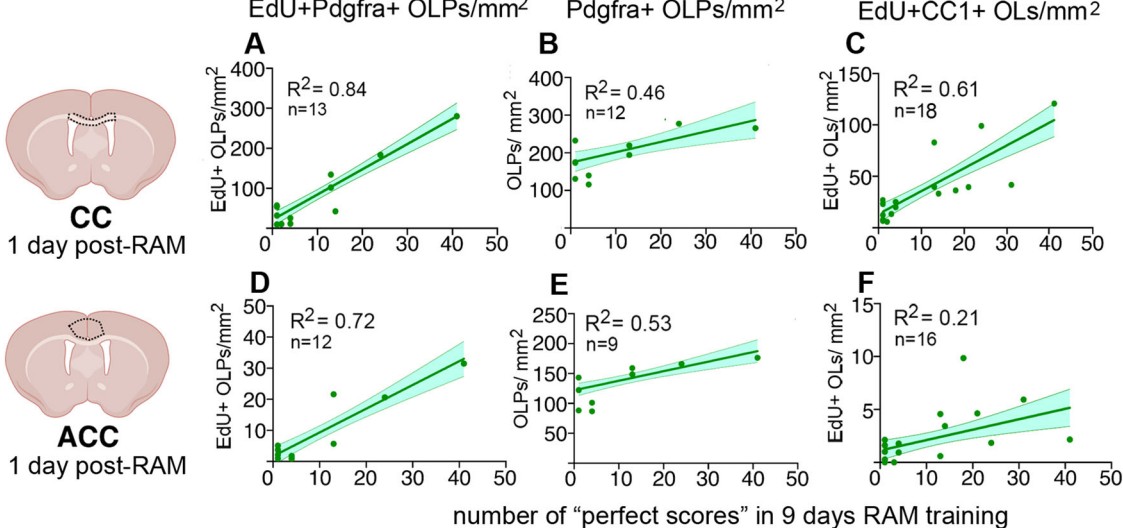

number of "perfect scores" in 9 days RAM training

**Fig. 4 | Working memory performance correlates with training-induced OLP proliferation and differentiation in individual mice.** The working memory performance of individual mice in the RAM (estimated by number of "perfect trials" during the 9 days of RAM training) correlates closely ($R^2 > 0.6$) with the number-density of proliferating OLPs (Pdgfra⁺EdU⁺) counted on 1-day post-training in either their corpus callosum (CC) (**A**) or anterior cingulate cortex (ACC) (**D**), and with the density of newly-generated OLs (CC1⁺ EdU⁺) in the CC (**C**). Significant correlations

($R^2 - 0.5$) were also observed between performance and OLP population densities (**B, E**). Lines of best fit (simple linear, least-squares regression) are drawn with 95% confidence intervals; $R^2$ and n values are shown on graphs and in Supplementary Data 3, together with slopes and intercepts. Corresponding data for prelimbic/ infralimbic cortex, mediodorsal thalamus, hippocampal CA1 and fimbria are shown in Supplementary Fig. S4. Source data are provided as a Source Data file. Drawings were created using BioRender.

to poor-performing controls), or from the almost complete failure of new OL generation (even lower than poor-performers).

## Working memory performance is proportional to OLP proliferation and differentiation

It was striking that good-performing mice generated many more new OLs, on average, than their poor-performing counterparts. It was also noticeable that there was generally a wider spread of data points among the good-performers than among the poor-performing mice (e.g. Figure 3C, D) – raising the possibility that, even among the good-performers, different behavioural outcomes of RAM training might reflect different levels of training-induced OLP proliferation and differentiation. We tested this by plotting the number of "perfect scores" achieved by individual mice (both good- and poor-performers) against the numbers of newly divided EdU⁺ Pdgfra⁺ OLPs and new EdU⁺ CC1⁺ OLs present in different brain regions post-training. Data for the anterior CC and ACC are shown in Fig. 4 and for PLC/ILC, MDT, CA1 and fimbria in Supplementary Fig. S3 (statistics in Supplementary Data 3). Strikingly, at 1-day post-RAM in the anterior CC there were strong correlations between the RAM performance of individual mice and the

number of EdU⁺ Pdgfra⁺ recently-divided OLPs ($R^2 = 0.84$), the overall population density of Pdgfra⁺ OLPs ($R^2 = 0.46$) and the number of EdU⁺ CC1⁺ newly-generated OLs ($R^2 = 0.61$) (Fig. 4A-C).

In another white matter tract, the fimbria, there was also a high correlation at 1-day post-RAM between working memory performance and the number-density of EdU⁺ Pdgfra⁺ recently-divided OLPs ($R^2 = 0.76$) but no significant correlations with either Pdgfra⁺ OLP population density ($R^2 = 0.14$) or density of EdU⁺ CC1⁺ newly-formed OLs ($R^2 = 0.28$) (Supplementary Fig. S3G-I and Supplementary Data 3).

In the ACC at 1-day post-RAM we found strong correlations between individual RAM performance and the number of EdU⁺ Pdgfra⁺ OLPs ($R^2 = 0.72$) and Pdgfra⁺ OLP population density ($R^2 = 0.53$) (Fig. 4D, E). There was also a weak correlation between RAM performance and number of EdU⁺ CC1⁺ newly-generated OLs ($R^2 = 0.21$) (Fig. 4F). Similar high correlations between working memory performance and density of EdU⁺ Pdgfra⁺ proliferating OLPs were observed in the PLC/ILC, MDT and hippocampal CA1 ($R^2 - 0.7$ in all three regions) (Supplementary Fig. S3A,D and Supplementary Data 3).

The behavioural experiments described above were conducted exclusively with male mice, to avoid potential variation among females

at different stages of the estrus cycle, or induced in males by proximity to them. Nevertheless, we repeated some of our experiments using female mice. Female and male mice achieved similar success scores during RAM training (day 9 scores: males 78% ± 2.3% vs females 85% ± 3.2%, means ± s.e.m., $p = 0.7$, 2-way ANOVA with Šídák's post-test) with similar numbers of perfect trials (day 9: males 31% ± 5.4% vs females 45% ± 11.3%, $p = 0.7$) (Supplementary Fig. S1M). Moreover, like males, female good-performers developed increased number-densities of proliferating OLPs, total OLPs and newly generated OLs relative to poor-performers during RAM training (Supplementary Fig. S1N-P). Therefore, we believe that the results and conclusions of our present study are likely to apply to males and females equally.

### Training-induced myelination of axons

The marked and persistent increase in numbers of newly-generated OLs in task-relevant white matter tracts (anterior corpus callosum and fimbria) in good-performing RAM-trained mice raised the question of whether these new OLs formed additional myelin sheaths. Additional sheaths are necessarily accompanied by extra nodes of Ranvier and paranodal loops, so we asked whether the number-density of node/paranode structures increased in the corpus callosum (CC) following successful RAM training. We immunolabelled sections of CC for voltage-gated sodium channel 1.6 ($Na_V1.6$, present in the axonal membrane at nodes) and the adhesion molecule Caspr (in the axonal membrane under the paranodal myelin membrane loops) (Fig. 5A), at 1-day post-training. We counted node/paranode structures in coronal sections through the anterior CC, which carries transverse connections between left and right ACC, of good-performing RAM-trained mice and home cage controls. This analysis suggested that there might be an increase in the density of nodal structures – hence myelin sheaths – in RAM-trained mice, although this did not reach statistical significance [control, 6.44 ± 0.36 nodes/100 $\mu m^2$, $n = 7$ mice; RAM-trained, 7.48 ± 0.36 nodes/100 $\mu m^2$, $n = 7$ mice, means ± s.e.m., $p = 0.066$, t = 2.03, df=12 (unpaired Student's 2-tailed t-test)] (Fig. 5B). We also measured the lengths of nodes and paranodes in these immunolabelled sections and found that, at this level of analysis, there was no detectable change in the average lengths of nodes, paranodes or complete node/paranode structure in RAM-trained mice versus home-cage controls (Fig. 5C-E). [Nodes: control, 1.69 ± 0.20 μm, $n = 780$ from 4 mice; RAM-trained, 1.51 ± 0.11 μm, $n = 737$ from 4 mice, p = 0.44, t = 0.82, df=6. Paranodes: control, 1.57 ± 0.05 μm, $n = 1106$ from 4 mice; RAM-trained, 1.47 ± 0.03 μm, $n = 1084$ from 4 mice, $p = 0.17$, t = 1.6, df=6. Complete nodal structures: control, 4.79 ± 0.27 μm, $n = 534$ from 4 mice; RAM-trained, 4.47 ± 0.06 μm, $n = 542$ from 4 mice, $p = 0.32$, t = 1.1, df=6 (unpaired Student's 2-tailed t-tests)].

The distribution of nodal structures was non-uniform in the CC (Fig. 5A) suggesting that sampling variation might obscure otherwise significant effects. We therefore turned to electron microscopy (EM), hoping for more reproducible selection of target areas, especially in the anterior-posterior dimension. Back-scatter EM from the surfaces of parasagittal sections mounted on glass microscope slides (Methods) allowed us to observe and image large uninterrupted areas of tissue. We counted all end-on profiles of myelin internodes, nodes and paranodes in ~50 μm-wide strips spanning the full ~300 μm dorsal-ventral extent of the CC, 600-650 μm from its anterior tip (Fig. 5F, G). We compared good-performing RAM-trained mice with home cage controls in this experiment so as to be able more easily to match numbers of trained vs control mice and since we had previously found no differences between numbers of new OLs induced by RAM-training in poor-performers versus home cage controls (Fig. 3). We found that there was possibly a small but non-significant (~10%) increase in the number of myelinated axon profiles in good-performing RAM-trained mice analyzed at 1-day post-training versus controls (good performers: 125 ± 6 /mm², 14,231 myelinated axon profiles counted from $n = 6$ mice; home cage: 113 ± 6 /mm², 12,871 myelinated axons from $n = 6$ mice,

means ± s.e.m., unpaired Student's 2-tailed t-test $p = 0.16$, t = 1.4, df=10) (Fig. 5H), accompanied by a significant ~25% increase in the number of node/paranode profiles (good performers: 6.1 ± 0.33 /mm², 689 nodes/paranodes counted from n = 6 mice; home cage: 4.8 ± 0.29 /mm², 545 nodes/paranodes from $n = 6$ mice, unpaired Student's 2-tailed t-test $p = 0.016$, t = 2.9, df=10) (Fig. 5I), consistent with an increase in the number of discrete myelin internodes. This indicates that the new training-induced EdU⁺ CC1⁺ OLs form new myelin sheaths – although we cannot rule out that pre-existing OLs are partly responsible for the additional myelin internodes. In RAM-trained animals the ratio of node/paranode profiles to all myelinated axon profiles ("node+paranode frequency") was also increased (good performer: 4.8% ± 0.15%, $n = 6$ mice; home cage: 4.3% ± 0.22%, $n = 6$, unpaired Student's 2-tailed t-test $p = 0.050$, t = 2.2, df=10) (Fig. 5J), suggesting that the extra myelin sheaths might be shorter than the majority, on average, in keeping with those extra sheaths being newly-formed. The alternative interpretation, that nodes/paranodes might be longer, on average, in RAM-trained versus control mice, was ruled out by direct measurement of nodes (previous paragraph, Fig. 5C-E).

As an alternative approach for detecting newly-forming myelin sheaths we used a tamoxifen-inducible reporter, *Tau-mGFP*, that expresses a membrane-associated GFP that reveals whole cell morphology. We generated *Pdgfra-CreER^T2^: Tau-mGFP* double-transgenic mice to visualize newly generated OLs (from differentiating *Pdgfra*-expressing OLPs) following tamoxifen administration. We administered a lower than usual dose of tamoxifen (200 mg/kg body weight) for two days immediately prior to diet restriction, provided EdU in the drinking water during the 9 days of RAM training/testing and perfusion-fixed the mice 1-day post-training (Fig. 6A). The lower tamoxifen dose lessened ill effects on the mice (e.g. weight loss, lethargy), allowing us to dispense with our usual 3-week recovery period prior to starting dietary restriction. The recombination efficiency of the *Tau-mGFP* reporter was also much reduced, allowing us to visualize individual, widely separated OLs.

Using this approach we detected multiple mGFP⁺ OLs with myelinating morphology in both the ACC (Fig. 6B, G) and the anterior CC (Fig. 6C, J). The great majority of these were positive for CC1 antigen and were a mixture of EdU⁺ and EdU-negative OLs (Fig. B',C'). The number-density of mGFP⁺ myelinating OLs was strongly increased in the corpus callosum of good-performing RAM-trained mice compared to home cage controls (Fig. 6D, E). The mGFP⁺, EdU-negative OLs (~80% of all mGFP⁺ OLs) must have formed prior to RAM training (during habituation or dietary restriction, before EdU was administered) or else during training from OLPs that did not divide before differentiating ("direct" OLP differentiation) (Fig. 6D: good performers, 120 ± 35 OLs/mm², $n = 3$; home cage controls, 17 ± 3 OLs/mm², $n = 3$. Means ± s.e.m., $p = 0.044$, t = 2.9, df=4). The mGFP⁺, EdU⁺ OLs (~20% of all mGFP⁺ OLs) must have formed from OLPs that divided during training when EdU was present (Fig. 6E: good performers, 26 ± 6 OLs/mm², $n = 3$; home cage controls, 1 ± 0.1 OL/mm², $n = 3$. Means ± s.e.m., $p = 0.012$, t = 4.4, df=4). GFP⁺ EdU⁺ CC1-negative OLs were rare (Fig. 6F: good performers, 2 ± 0.8 OLs/mm², $n = 3$; home cage controls, 3 ± 0.9 OLs/mm², $n = 3$), so we conclude that many of the additional CC1⁺ OLs that form during RAM training (e.g. Figure 3) most likely synthesize new myelin in both the white and gray matter.

In the ACC at 1-day post-training, many of the new myelin sheaths were laid down on previously unmyelinated regions of axons, judging by the fact that many GFP⁺ internodes terminated at a heminode at one or both ends. These heminodes, marked by an island of Caspr immunolabelling that overlapped with mGFP, were not associated with clustered sodium channels ($Na_V$) so presumably were unable to support saltatory conduction and acceleration of action potential propagation (arrowheads in Fig. 6H, I). However, some of the new internodes (around 20% in the OL illustrated in Fig. 6G) were laid down next to pre-existing internodes, because they terminated in normal-appearing

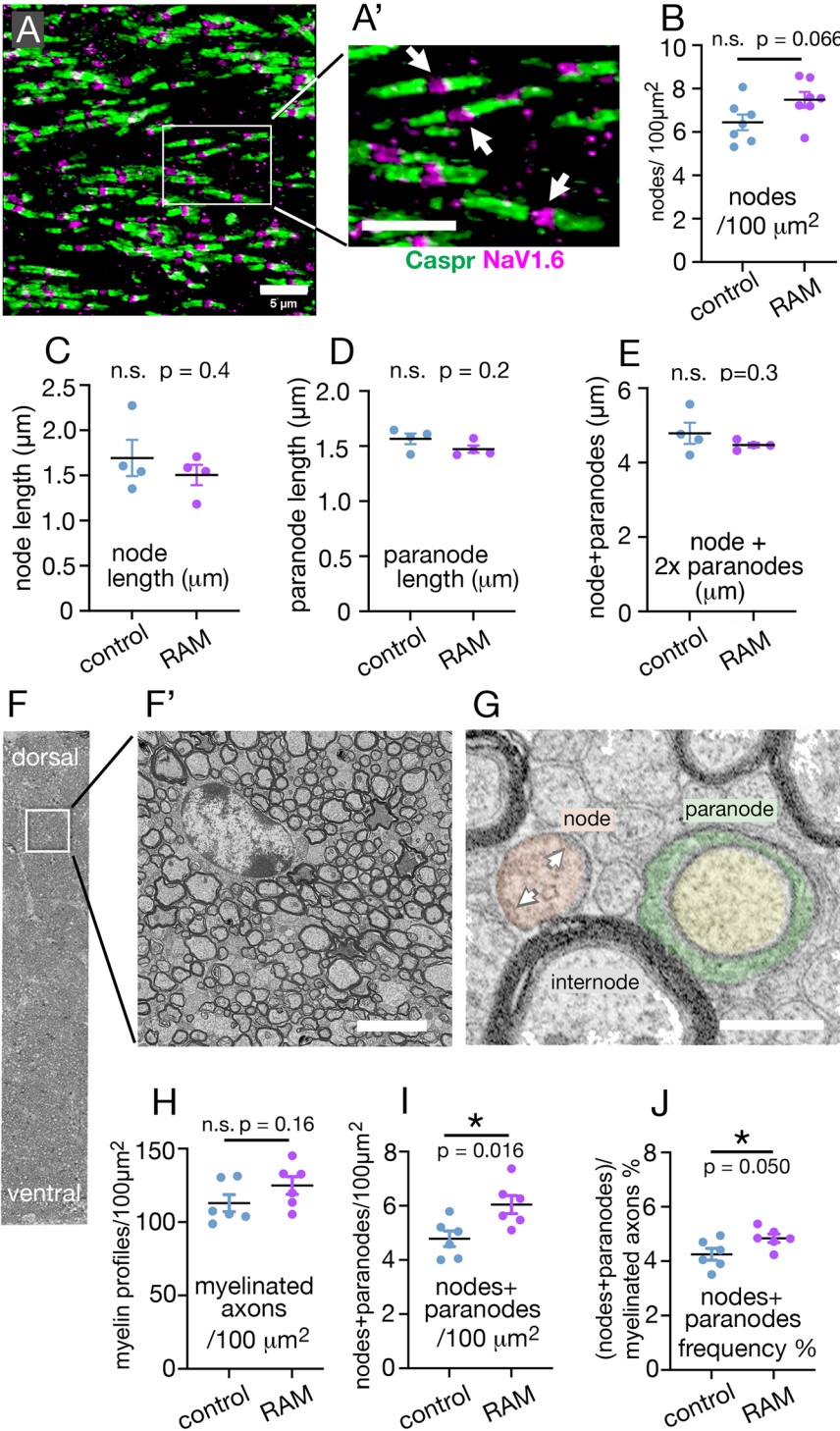

nodes of Ranvier comprising a cluster of Na$_v$1.6 flanked by two islands of Caspr (arrows in Fig. 6H, I). These new internodes are likely to accelerate action potentials, although whether this is the primary role of new OLs in working memory training remains conjectural.

In the white matter of the CC it was more difficult to characterize nodal structures because of their high density and because of the usually inferior fixation and immunolabelling achieved in white matter. Nonetheless, some of the new myelin sheaths terminated in what appeared to be mature nodes with clustered Caspr and Na$_v$1.6 (Fig. 6K), while others, as in gray matter, apparently terminated in hemi-nodes (Fig. 6L).

## No positive correlation between working memory performance and bulk neuronal activity

Improvement in working memory performance undoubtedly reflects altered activity of neural circuits that are engaged in the task, but how circuit activity and OL dynamics interact is not known. As a first step towards understanding the relationship between neuronal activity and OL dynamics we performed c-Fos immunolabelling on sections of ACC, where we had observed the most pronounced grey matter effects on OL dynamics, immediately after RAM training.

We compared good-performers, poor-performers and home cage controls that were perfusion-fixed within one hour following the final

**Fig. 5 | Working memory training stimulates new myelin sheath production.**
**A** Confocal images of coronal sections through anterior corpus callosum (CC) at the level of the anterior cingulate cortex (ACC) were immunolabelled for Na$_v$1.6 (magenta) and Caspr (green) to visualize nodes and paranodes respectively. (**A'**) is a higher-magnification view of the area indicated in (**A**); *arrows* indicate nodes of Ranvier. These micrographs are representative of more than 3 independent immunolabelling experiments. **B** Node/paranode structures were counted in photographic images of 5 μm confocal stacks; there is a trend towards higher node/ paranode density in good-performing RAM-trained mice (*n* = 7) compared to home-cage controls (*n* = 7). **C**–**E** We measured the lengths of mature nodes of Ranvier as in Arancibia-Cárcamo et al. [86] (see Methods). The mean lengths of nodes (**C**), paranodes (**D**) and complete nodal structures (node flanked by two paranodes, **E**) were all unchanged in RAM-trained mice (n = 4) compared to home-cage controls (*n* = 4). **F** Part of a wide-field EM backscatter image (parasagittal), including the entire dorsal-ventral extent of the CC, 600 μm from its anterior tip; (**F'**) is a higher-magnification image of the area indicated in (**F**). **G** EM profiles of myelinated fibres

sectioned through a node (false-colour orange), a paranode (false-colour green/ yellow) and a myelin internode. Nodes can be distinguished from unmyelinated axons by the presence of electron-dense material undercoating the axonal membrane (*arrows*). All myelin internode (**M**), node (**N**) and paranode (**P**) profiles were counted in single - 260 ×50 μm areas of 100 nm thick sections such as (**F**) for each individual RAM-trained or home-cage control mouse (*n* = 6 of each). **H** There was no significant change in myelinated axon (M + N + P) density in RAM-trained mice (*n* = 6) versus controls (*n* = 6), but there was a significant increase in the combined N + P density (**I**) and a marginally significant increase in the ratio (N + P)/(M + N + P) ("nodes + paranodes frequency") (**J**). Together these data indicate that there are more nodal structures, hence more internodes in RAM-trained versus control mice and suggest that those extra internodes might be shorter than the majority, consistent with their being recently-formed. Data are presented as mean ± s.e.m. (Student's two-tailed t-test). n.s. not significant ($p > 0.05$), * $p \leq 0.05$, actual p-values specified in the Figure. Source data are provided as a Source Data file. *Scale bars:* **A**-**A'**, 5 μm; **F'**, 5 μm; **G**, 2 μm.

trial on day 9 of training (Fig. 7A-C). EdU was administered to the mice throughout training. c-Fos expression generally remains elevated in activated neurons for ~2 hours, so this approach should capture neurons that were active during the final 3-4 trials of the final training day. c-Fos activity in the ACC was assessed in two ways: 1) by estimating the number-density of neurons that expressed c-Fos immunoreactivity above background levels and 2) by measuring the fluorescence intensity of c-Fos$^+$ neuronal nuclei (see Methods). We found that both these measures tended to be reduced in good-performers and more tightly clustered around the mean, relative to either poor-performers or home cage controls (Fig. 7D, E), although the differences did not reach statistical significance (p > 0.05; Kruskal-Wallis non-parametric statistics; Fig. 7 legend and Supplementary Data 4). There was no positive correlation between working memory performance and the average c-Fos fluorescence intensity per neuron ($R^2 = 0.056$; Fig. 7F). There was a weak *negative* correlation between performance and the number of c-Fos$^+$ neurons ($R^2 = 0.21$; Fig. 7G).

To confirm that OL dynamics had been stimulated in this experiment we also counted EdU$^+$, Pdgfra$^+$ dividing OLPs and EdU$^+$, CC1$^+$ new OLs in the ACC and underlying corpus callosum (CC). As expected, the number-densities of both EdU$^+$ OLPs and EdU$^+$ newly-formed OLs were increased in good-performers compared to both poor-performers and home cage controls. The working memory performance of the trained mice correlated positively with OL lineage metrics ($R^2 = 0.43$ in ACC and $R^2 = 0.68$ in CC for EdU$^+$ OLPs; $R^2 = 0.27$ in both ACC and CC for EdU$^+$ OLs) (Fig. 7H-K). When we plotted the number-density of c-Fos$^+$ neurons directly against the densities of either EdU$^+$, Pdgfra$^+$ OLPs or EdU$^+$, CC1$^+$ OLs in the ACC and CC, there were no obvious correlations (Fig. 7L-O). These data indicate that the relationship between OL dynamics and population-level neuronal activity is neither simple nor direct.

## Discussion

Experimental blockade of OL genesis in adult *Myrf*-cKO mice has revealed that active production of new myelinating OLs supports motor skill learning and memory and the consolidation of remote (long-term) fear and spatial memories[10–13]. We have now shown that active OL genesis is required for mice to improve their performance with training on T-maze and 8-arm radial maze tasks that exercise and assess spatial working memory – also known as delayed non-matching to position (DNMP) or win-shift behaviour[32,37]. *Myrf*-cKO mice did not improve their performance in these maze tasks over 8-9 days of post-habituation training – unlike their control littermates, which improved steadily during the training period. This demonstrates that *Myrf*-cKO mice were unable to improve their spatial working memory performance. The requirement for OL genesis in working memory training is intriguing because working memory capacity is known to underpin all

kinds of cognitive abilities and correlates closely with measures of intelligence in humans and animals[38–41].

Strikingly, the performance of individual mice in our radial arm maze task correlated strongly with the numbers of additional OLPs and OLs generated by division and differentiation of OLPs during training. Experiments with the *Tau-mGFP* reporter demonstrated that many of the newly-formed OLs in cortical grey matter at 1-day post-training formed new myelin sheaths. However, only a fraction of the new sheaths in gray matter formed normal-appearing nodes of Ranvier with clustered Na$_v$1.6, the majority forming heminodes that did not cluster Na$_v$1.6. It is perhaps unlikely that these heminodes contribute to accelerated conduction velocity but they and their associated internodes might provide other functional input to the axons of both local interneurons and long-range output neurons. Most of the additional OLs that form in the ACC of good-performing animals are eliminated in the 14 days post-training, so their influence on the neuronal circuitry is presumably transient.

We also visualized many newly-formed (mGFP$^+$) OLs in the sub-cortical white matter (CC) that were synthesizing myelin at 1-day post-training. The functional maturity of their myelin sheaths was difficult to assess, but some of them terminated in normal-appearing nodes, while others terminated at heminodes, like their counterparts in gray matter. It is possible that heminodes mature into full nodes as their myelin sheaths elongate, gather Na$_v$ channels and collide with other, pre-existing heminodes. Further experiments will be required to investigate this. The fact that heminodes are observed at all suggests that the sparse pattern of myelination observed on the axons of projection neurons in the cortex[42] might be maintained as those axons traverse the corpus callosum.

Most of the extra CC1$^+$ OLs that formed in the CC during training survived for at least 14 days post-training (Fig. 3D, G). It will be interesting to discover whether all the myelin sheaths of these surviving OLs assemble mature, functional nodes and whether their survival lifetime correlates with the lifetime of training-induced working memory improvement.

We previously found evidence for two waves of OL generation, before and after OLP division, during motor skills learning[10,11]. We speculated that direct OL differentiation temporarily depletes the OLP pool, which is then kicked into cell division by a homeostatic mechanism − e.g. through a temporary glut of mitogenic growth factors such as PDGF[43] or by loss of contact inhibition[44] − in order to replenish the OLP population. It is possible that similar OL dynamics occurs during RAM training, but the homeostatic replacement of OLPs overshoots, creating an excess of OLPs that might accelerate or prolong production of OLs in the second wave. Improvement of working memory then might depend on direct OL differentiation, while OLP division and secondary OL differentiation consolidate the

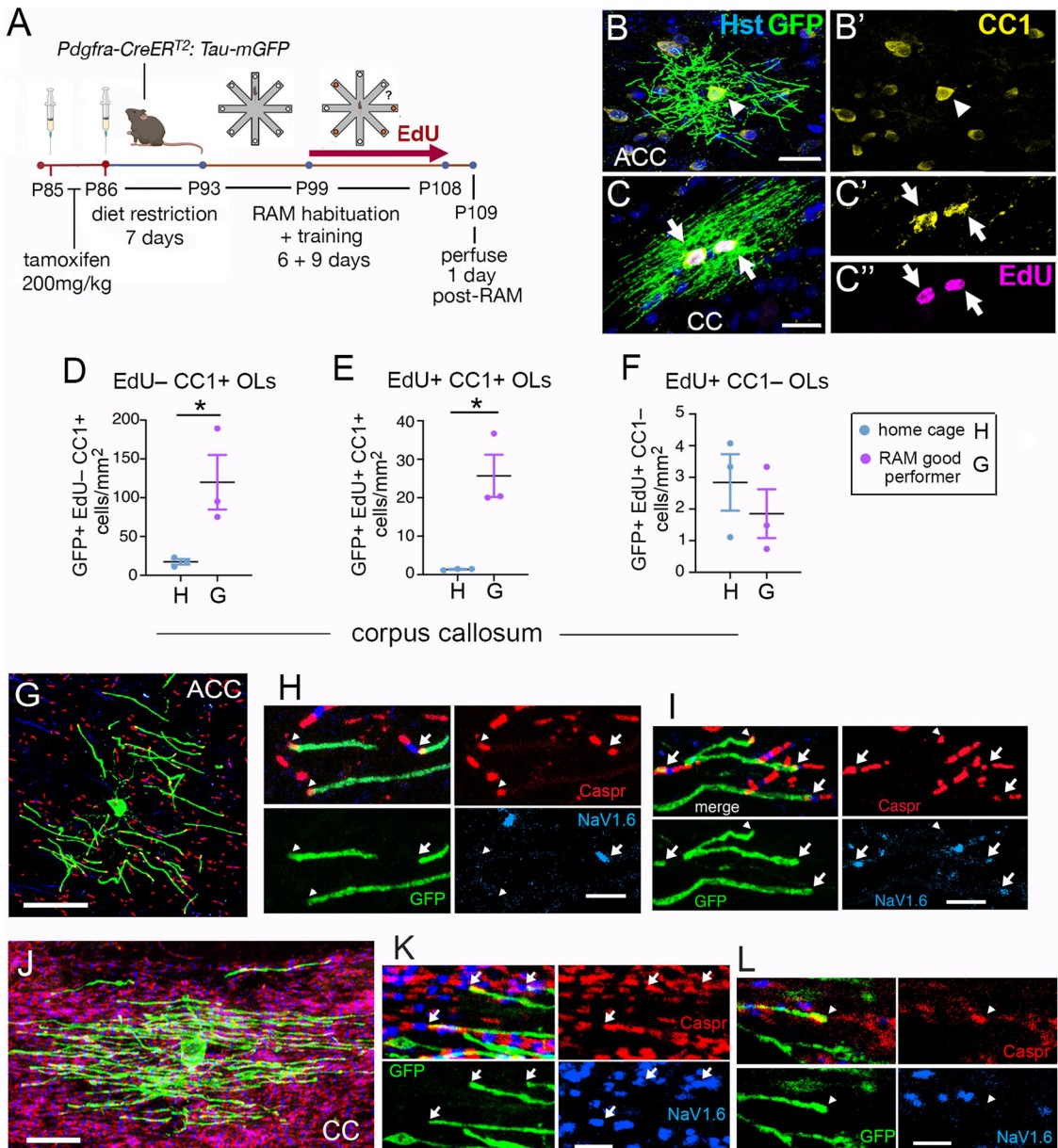

**Fig. 6 | Working memory training stimulates production of myelin-forming OLs. A** Experimental protocol (drawing created using BioRender). *Pdgfra-CreER^{T2}: Tau-mGFP* mice were given a lower than normal dose of tamoxifen (200 mg/kg on 2 consecutive days) immediately before dietary restriction followed by RAM habituation and training as before. EdU was given in the drinking water during RAM training and the mice were perfusion-fixed 1-day post-training. mGFP-expressing OLs (generated from Pdgfra⁺ OLPs post-tamoxifen) were double-immunolabelled with anti-GFP and monoclonal CC1, followed by EdU chemistry (**B**, **C**). Newly-formed GFP⁺ OLs with myelinating morphology were visualized in both the anterior cingulate cortex (ACC) (**B**) and anterior corpus callosum (CC) (**C**). The majority of GFP⁺ OLs was also CC1⁺ (**B,C**). Both EdU⁺ OLs (*arrows*) and EdU-negative OLs (*arrowheads*) were observed. **D, E** Number densities of GFP⁺ EdU⁻ CC1⁺ and GFP⁺ EdU⁺ CC1⁺ OLs with myelinating morphology were greatly increased in the corpus callosum of good RAM-performers ($n = 3$) relative to home cage controls ($n = 3$) [EdU⁻ OLs: $120 \pm 35$ OLs/mm² vs $17 \pm 3$ OLs/mm² in good-performers vs controls ($p = 0.012$). EdU⁺ OLs: $26 \pm 6$ OLs/mm² vs $1.4 \pm 0.12$ OLs/mm² in good-performers vs controls ($p = 0.044$). means ± s.e.m., Student's two-tailed t-tests]. **F** Small numbers of GFP⁺ EdU⁺ CC1⁻ OLs (presumably pre-myelinating) were also observed in the corpus callosum of both good RAM-performers and home cage controls ($n = 3$ for both). **G-L** Na$_V$1.6 and Caspr immunolabelling showed that some GFP⁺ new myelin sheaths in both ACC and CC terminated in normal-appearing paranode/node structures with clustered Na$_V$1.6 (*arrows*), while others terminated in heminodes that did not appear to cluster Na$_V$1.6 (*arrowheads*) ($n = 3$). Micrographs are representative of more than 3 independent immunolabelling experiments. Data are presented as mean ± s.e.m. Source data are provided as a Source Data file. *Scale bars*: 20 μm (**B, C, G, J**), 2 μm (**H, I, K, L**).

improvement. Alternatively, or in addition, enhanced OLP proliferation and the consequent increase in OLP population density, which persisted for at least 2 weeks post-training in the white matter, might itself lead to improved cognitive performance. There are reports that immature OL lineage cells in zebrafish[45,46] and mice[47–49] can exert effects on neurons that are unrelated to myelination, including synaptic pruning by OLPs[47–49]. It is conceivable that such non-canonical

effects could have a lasting impact on neural circuitry even if the responsible OL lineage cells are eventually eliminated.

The dynamic changes to OL lineage cells and myelin that we observe are very likely triggered directly or indirectly by changes in circuit activity as the animals learn. The newly-generated OL lineage cells must in turn modify the behaviour and activity of the circuits with which they interact, so that learning and memory formation is a

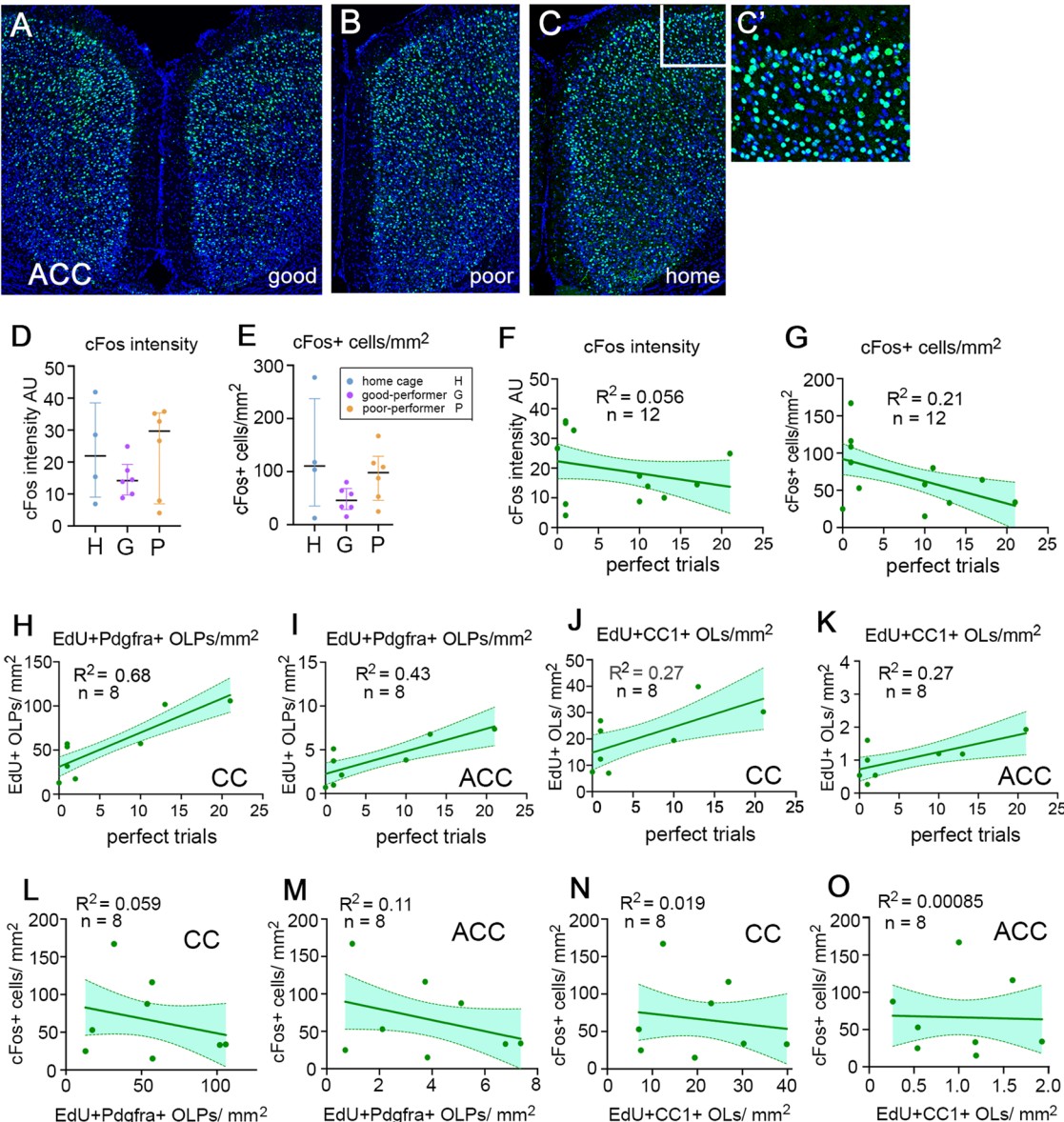

**Fig. 7 | Working memory performance and c-Fos immunoreactivity in the mPFC. A–C** c-Fos immunolabelling in the ACC of good- and poor-performing mice ($n = 6$ for both) (**A**, **B**), perfusion-fixed immediately following 9 days of RAM training, alongside a home-cage control ($n = 4$) (**C**). **C'** is a higher-magnification view of the area outlined in (**C**). **D** Average fluorescence intensities (arbitrary units, A.U.) and (**E**) number-densities (cells/mm²) of c-Fos⁺ neuronal nuclei in the ACC of good- and poor-performers and home cage controls (median and 25%-75% interquartile range (IQR) is shown). [Kruskal-Wallis non-parametric statistics: c-Fos intensities (arbitrary units), median (IQR): good 14 (10-19); poor 30 (7-35), $q = 0.61$, $p = 0.33$; home cage 22 (9-39), $q = 0.61$, $p = 0.39$, H-statistic 1.2. number of c-Fos⁺ neurons per 100 μm², median (IQR): good 46 (29-68); poor 98 (46-129), $q = 0.23$, $p = 0.15$; home cage 111 (35–238), $q = 0.23$, $p = 0.13$, H-statistic 3.1.]. **F**, **G** The working memory performances of individual mice in the RAM (estimated by number of "perfect

trials" during the 9 days of RAM training) were plotted against the mean fluorescence intensities (**F**) or number-densities (**G**) of c-Fos⁺ neuronal nuclei. Lines of best fit (simple linear, least-squares regression) are drawn with 95% confidence intervals; $R^2$ and n values are shown. There was a weak inverse correlation between RAM performance and the density of c-Fos⁺ neurons in the ACC ($R^2 = 0.21$). **H–K** In the same mice there was a strong positive correlation between RAM performance and OLP proliferation in the CC and the ACC ($R^2 = 0.68$ and 0.43, respectively) and weaker correlations with the density of newly-generated OLs in both regions ($R^2 = 0.27$ in both CC and ACC). **L–O** There were no obvious correlations between the number of c-Fos⁺ neurons in the ACC and the numbers of either newly-divided EdU⁺, Pdgfra⁺ OLPs (**L**, **M**) or newly-differentiated EdU⁺, CC1⁺ OLs (**N**, **O**) in the ACC or the adjacent CC. Full Kruskal-Wallis statistics are given in Supplementary Data 4. Source data are provided as a Source Data file.

reciprocal dialogue between neurons and OL lineage cells – potentially even extending to other cells such as astrocytes, microglia and peri-vascular cells. We attempted to detect differences in the level of activity of ACC neurons in good- versus poor-performers or home cage controls, by immunolabelling for the immediate-early gene c-Fos. Intriguingly – and against expectation – there was a trend towards a lower number and intensity of c-Fos⁺ neurons in good-performers versus controls, and a tighter clustering around the mean, although this did not reach statistical significance (Kruskal-Wallis non-

parametric analysis). There was even a weak negative correlation between the number of c-Fos⁺ neurons and working memory performance. These observations invite speculation that one outcome of altered OL dynamics might be to limit activity to a reduced subset of task-relevant neurons, potentially cutting out background noise and allowing more channeled and efficient problem-solving behaviour in good- versus poor-performers.

The involvement of myelin in human cognition and particularly working memory has been suspected for some time[50]. For example,

the spatial working memory performance of individual pre-adolescent children (ages 3-13) was found to correlate with fractional anisotropy (FA) in the superior longitudinal fasciculus, a white matter tract underlying the dorsal PFC, independent of chronological age[51]. FA is an MRI measure of the directional dependence of tissue diffusivity, believed to reflect, partly, the extent of myelination of aligned axons in white matter tracts. FA was also shown to increase (suggesting increased myelination) in task-relevant white matter tracts of young adults undertaking working memory training with the N-back test[15]. However, FA is at best an indirect measure of myelination; the histological and behavioural experiments we report here are the first unequivocal demonstration of a requirement for adaptive OL dynamics during working memory tasks in mice. Thus, improvements in working memory performance with training resembles motor skills training, in that both require reiterative practice over days and both rely on OL dynamics and presumably neo-myelination.

How might OL genesis and neo-myelination improve working memory performance? Neuronal activity persists ("reverberates") within and among the pre-frontal cortex, parietal cortex and other brain areas (e.g. mediodorsal thalamus) for a limited period after the initial stimulus has ceased (the "delay" period). One long-held idea is that this reverberant activity maintains information in working memory and that the power and decay time of the reverberations determine working memory capacity and duration[52–56]. This general idea has been challenged by the discovery of activity-silent or latent working memory traces, held at the level of modified synapses[57–61]. Despite this, it seems likely that both persistent neuronal activity and synaptic modification can co-contribute to working memory maintenance during the delay period[55,62,63]. If so, an expected outcome of working memory training might be to reinforce or prolong reverberatory activity, for which there is some evidence in humans[64]. It is conceivable that reverberation is limited by the energy required and available to maintain circuit activity, in which case myelination might improve working memory by 1) reducing the energy needed to propagate action potentials and sustain reverberation, and 2) facilitating energy production within axons by providing them with metabolic substrates such as lactate[65].

Complex decision-making behaviours, including those that rely on working memory, are associated with rhythmic electrical activity in multiple brain regions and particularly the coordination and coherence of rhythmicity among different regions[55,66]. Many studies have shown that increased coordination of rhythmic activities correlates with behavioural outcomes. For example, in an "H-maze" test of spatial working memory, phase synchrony between theta-rhythms ( ~ 4-12 Hz) in hippocampal CA1 and the medial prefrontal cortex (mPFC) increased when rats approached a choice-point of the maze prior to making a correct choice of goal-arm, but not prior to making an incorrect choice[67]. This long-range theta coherence resulted in coordinated firing of neurons in CA1 and mPFC, as a result of individual neurons firing at specific phases of the theta cycle in both regions (spike-phase locking). Theta-coherence between hippocampal and mPFC neurons was also observed in mice during the choice phase of a T-maze DNMP task[68]. Moreover, theta-coherence increased during working memory training in the T-maze and, for individual mice, theta-coherence prior to training was predictive of subsequent learning ability[68]. More recently, enhanced phase coupling between different frequency bands − between theta and beta ( ~ 12-30 Hz) or theta and gamma ( ~ 30-120 Hz) − has been linked to the accuracy of working memory-based decisions of mice in a T-maze DNMP task[69,70]. Similar cross-frequency phase coupling correlates with working memory performance in primates, including humans (reviewed in ref. 55).

There is growing awareness that adaptive OL genesis and neo-myelination, by adjusting axonal conduction speeds, might play a key role in modulating and coordinating the sorts of rhythmic behaviours discussed above[13,71,72]. Simply put, faster communication between, say,

hippocampus and PFC would be expected to allow hippocampal output to drive prefrontal circuits to oscillate more closely in tune with those in the hippocampus, or vice versa. It is not so obvious how adaptive myelination might drive the rapid changes in phase relationships that are observed during the execution of a behavioural task − as mice approach and pass a choice point in a maze, for example. However, this could be explained by positing that different components of a working memory task − a temporal sequence of events, for example − are represented by distinct non-overlapping assemblies of neurons in the PFC, that only one assembly is active at a given time and that the shifting oscillatory relationships between PFC and hippocampus result from rapid switching between the different assemblies[73]. A recent live imaging study has demonstrated this organizational principle in macaques[74]. It follows that the phase relationships between hippocampus and distinct PFC neuron assemblies could potentially be adjusted independently of one another, by adaptive myelination within the separate assemblies.

Working memory dysfunction and impaired inter-regional synchronicity are both features of common neurodegenerative disorders and psychiatric conditions including Alzheimer's and Parkinson's disease[75], schizophrenia[68], autism spectrum disorder[76], attention deficit/ hyperactivity disorder[77] and others (reviewed in refs. 55,78). There are also consistent reports of links between myelin dysregulation and schizophrenia (reviewed in refs. 79,80). Hence, how adaptive myelination feeds into normal working memory and cognition, how disrupting those processes can lead to psychiatric disorders and how, in the longer term, this information might lead to benefits for affected individuals are important questions for the future.

## Methods

### Mice

Mouse experiments were pre-approved by the UCL Animal Welfare and Ethical Research Board (AWERB) and conformed to the regulations laid down in the Animals (Scientific Procedures) Act 1986 and subsequent amendments, introduced and overseen by the UK Government. *Myrf* (flox/flox): *Pdgfra-CreER*[T2]: *Rosa26-YFP* mice were crossed to *Myrf* (flox/+): *Pdgfra-CreER*[T2]: *Rosa26-YFP* on a mixed C57BL/6, CBA, 129 genetic background – mainly C57BL/6, although coat colour was agouti, hence CBA-derived. *Myrf* flox mice were provided by Ben Emery[81]. *Pdgfra-CreER*[T2][82] and *Rosa26-YFP*[83] were homozygous throughout. This cross generates similar numbers of *Myrf* (flox/flox): *Pdgfra-CreER*[T2]: *Rosa26-YFP* (*Myrf*-cKO) and *Myrf* (flox/+): *Pdgfra-CreER*[T2]: *Rosa26-YFP* (control) littermates that can be distinguished by genotyping as previously described[10,11]. *Pdgfra-CreER*[T2] homozygosity was determined by qPCR using the following primers (5'-3'): TGACGGTGGGAGAATGTTAATC (Cre-f), GCTACACCAGA-GACGGAAATC (Cre-r), ATGACA TCAAGAAGGTGGTG (GAPDH-f), CATACCAGGAAATGAGCTTG (GAPDH-r). Genotyping was carried out after behavioural testing so that the experimenter was blind to genotype during testing. The one exception was the left-right discrimination task, in which it was necessary to genotype prior to testing in order to be able to counterbalance numbers of *Myrf*-cKO and control mice between left and right goal-arms.

*Tau-lox-STOP-lox-mGFP-IRES-lacZ* reporter mice[84,85] (referred to as *Tau-mGFP*, obtained from Sylvia Arber) were crossed with *PdgfraER*[T2] to generate double-transgenic offspring for visualization of whole cell morphology of newly-generated OLs following tamoxifen administration.

### Tamoxifen and EdU

Tamoxifen (Sigma) was prepared on the day of administration by dissolving at a concentration of 40 mg/ml in corn oil by sonicating for one hour at 20-37 °C in a sonicating water bath. Tamoxifen was administered by oral gavage at a dose of 300 mg/kg once a day on four consecutive days (P60-P63). Mice were left to recover for 3 weeks

before starting behavioural experiments. All mice undergoing behavioural tests, including the controls, received tamoxifen at the same time and dose. For sparse labelling of OLs in *Pdgfra-CreER*[T2]*: Tau-mGFP* mice, tamoxifen was administered at 200 mg/kg on two consecutive days.

To measure rates of cell generation, 5-ethynyl-2′-deoxyuridine (EdU) was administered via the drinking water at 0.2 mg/ml during T-maze or radial maze testing (8 or 9 days duration, respectively). For immunohistochemistry and EdU detection, mice were perfusion-fixed, then brains were removed under fixative and immersed overnight in 4% (w/v) paraformaldehyde in phosphate-buffered saline (PFA) on the final day of T-maze training, or either immediately after or one or 14 days after the final day of RAM training.

### Mouse behaviour

Mice were maintained on an artificial 12 h light-dark cycle. Room lights were turned on at 7am and the dark period commenced at 7 pm. Behavioural experiments were conducted between 9 am and 7 pm in a separate dedicated room. White noise at 75 dB was played throughout the day. Sessions were video-recorded for later analysis. For all experiments we used only behaviourally naïve male mice, which were group-housed at 2 to 5 mice per cage from weaning until the beginning of experiments.

### T-maze rewarded alternation

The delayed non-matching to position (DNMP) task was carried out as described[31,32,35] with modifications (Fig. 1 and Supplementary Video 1). At 11 weeks of age (P77), mice were caged singly and kept on a restricted diet of standard pellet chow to maintain them at a target ~85% of starting weight, to motivate them to seek food rewards. Along with a measured amount of standard chow we provided the mice with 1 ml of dilute sweetened condensed milk (Carnation, Nestle; 50% (v/v) in distilled water) in their home cage for eight days up to and including the first day of habituation, to familiarize them with the food reward used during the task. During this time the mice were handled daily to gain familiarity with the experimenter.

After moving to the behaviour room, mice were left undisturbed in their home cages for 5 min prior to the start of any procedures in the maze. The T-maze consisted of a start-arm and two goal-arms, left and right, with a manually-operated door in each arm close to the T-junction. There were 3 days of habituation. On day 1, mice were released in the start-arm, facing away from the T-junction, with all doors removed (no food rewards present), and allowed to explore the maze freely for 30 min. This was the last day that condensed milk was provided in the home cage. On day 2, mice were introduced into the left or right goal-arm (with food rewards present), all doors closed, and allowed to consume the reward (70 µl dilute condensed milk) before being transferred to the opposite goal-arm and again allowed to consume the reward. This was repeated for all mice in the cohort and the whole sequence repeated five times, so that each mouse received 5 left-arm and 5 right-arm rewards on the day. This procedure was repeated on day 3 but starting in the opposite goal-arm. Each experimental trial consisted of a "forced" and a "free" run separated by a 30-second interval (delay) in the home cage. Both goal-arms were baited before starting the trial. In the forced run, either the left or right goal-arm was closed; the mouse was released in the start-arm and allowed to enter and consume the reward in the one accessible goal-arm. In the free run, both goal-arms were open and the mouse had to enter the previously closed/unvisited arm in order to receive a second reward. After a correct arm choice, the mouse was allowed to consume the reward before being removed to its home cage. After an incorrect arm choice the mouse was confined in the arm for 15 s before being removed to its home cage. Each mouse in the cohort completed its trial in turn and the whole cohort undertook 10 trials per day for the 8 days of the experiment with a between-trials interval for a given mouse of

~40 min hour. There were 5 left- and 5 right-arm forced runs per mouse per day, no more than 3 consecutive forced runs on the same side. The threshold time for one trial was 5 minutes and mice that exceeded this were removed from the analysis. The performance score on each day was calculated as the number of correct choices per 10 trials x 100 (%).

### T-maze left-right discrimination

Experimentally naïve mice were assessed on their ability to acquire a simple left-right discrimination task using the same T-maze. This task places the same sensorimotor and motivational demands on mice but does not tax working memory. The apparatus and habituation steps were the same as for the rewarded alternation task. For each experimental trial, there was only a single run and only one goal-arm was baited (70 µl dilute condensed milk). For a given mouse, the same arm was baited in all trials; for the different mice in the cohort left and right arms were counterbalanced. At the start of the trial the start-arm door was closed and both goal-arm doors open. The mouse was released in the start-arm, then the door was opened and the mouse was allowed to enter one of the arms in search of a food reward. After a correct arm choice, the mouse was allowed to consume the reward before being removed to its home cage. After an incorrect arm choice, the mouse was confined in the unrewarded arm for 15 s before being removed to its home cage. Each mouse in the cohort was tested in turn, then this whole sequence of trials repeated 10 times per day for 3 days with an inter-trial interval of ~40 min for each mouse, while other mice were in the maze. In the reversal phase of the task, the identity of the rewarded goal-arm was switched from left to right (or vice versa) for each mouse, and the whole experimental procedure repeated for another 3 days. The performance on each day was calculated as the number of correct choices per 10 trials x 100 (%).

### Y-maze

A Y-maze was used to assess spontaneous spatial novelty preference in *Myrf*-cKO mice (*Pdgfra-CreER*[T2]*: Myrf*[flox/+], n = 11, *Pdgfra-CreER*[T2]*: Myrf*[flox/flox], n = 13). Tamoxifen was administered by oral gavage on 4 consecutive days P60-P64, and the test commenced 50 days later on P110. The test assesses rapidly acquired, short-term spatial memory and relies on the fact that normal mice prefer novel over familiar spatial environments. The Y-maze was constructed from transparent Perspex with 3 equally spaced radiating arms 30 cm×8 cm x 20 cm (length x width x height). The protocol was adapted from Bannerman et al.[86]. One of the arms was designated the"start" arm and the remaining arms the "other" and "novel" arms. The start arm was constant but the other and novel arms were allocated pseudo-randomly to each mouse, while counterbalancing arm selections within and between groups. During the first phase of the test (the exposure phase), the entrance to the "novel" arm was blocked with a sheet of opaque Perspex; a mouse was placed at the end of the start arm and allowed to explore the start and other arms for 5 min (beginning from the time the mouse first left the start arm). Entry into an arm was defined when the mouse placed all four paws into an arm. The mouse was removed from the maze to its home cage for 2 min, then returned to the end of the start arm and allowed to explore all 3 arms of the maze freely for 2 min (beginning from the time the mouse left the start arm). The time spent in each arm of the maze and the number of entries into each arm were recorded during both the exposure phase and the test phase. For the test phase, a discrimination ratio [novel arm/(novel + other arm)] was calculated for both arm entries and time spent in arms.

### 8-arm radial maze

Spatial working memory was also assessed using a semi-automated 8-arm radial maze, purchased from Tracksys Ltd (Nottingham, UK). The RAM consists of a central octagonal hub with eight radiating arms 27 cm × 6 cm × 15 cm (length × width × height), entry to which is controlled by individual servo-driven doors that open/ close vertically

from beneath the floor of the maze. The movement of mice in the maze was tracked by EthoVision XT13 video tracking and control software (Noldus, Wageningen, The Netherlands), programmed in-house (by T.S.) to operate the doors as required. The maze floor was illuminated evenly with indirect warm white light (about 80 lux at the centre) and white noise at ~75 dB was played during experiments. There were several distal visual cues such as a TV monitor and a metallic ball above the maze and abstract symbols (Zener cards) on the ceiling above the maze. The distance travelled, mean speeds and acceleration were calculated by EthoVision XT13 and the videos were edited using Blender 3.3.1.

The RAM protocol was based on the protocol described for rats by Sasaki et al.[33], adapted for mice by T.S. with input from S.N. The procedure and feeding regimen in the week before habituation were the same as for the T-maze rewarded alternation task. There were 6 days of habituation on the RAM, divided into 3 stages of 2 days each. Days 1 and 2 were to allow the mice to explore the maze and to become accustomed to the movement and sound of the doors. Individual mice were placed in the central hub and doors to all arms (unbaited) were opened after 5 s initial confinement and mouse tracking started. After the mouse entered an arm the door to that arm was closed for 3 s (the duration of up/down door movement) before re-opening; then after leaving the arm the mouse was confined in the central hub for 3 s before all doors were re-opened. This sequence was repeated for 30 min. The mouse was then returned to its home cage and the maze cleaned thoroughly. After day 2, condensed milk was no longer provided in the home cage. Days 3 and 4 introduced the mice to food rewards (70 μl of condensed milk diluted 1:1 with water), placed at the end of all arms in sunken wells so that they were not visible until the mice were within ~10 cm of the well. Mice were allowed to enter each arm at least twice without the doors operating to encourage free exploration (only the first visit was rewarded). Each mouse in the cohort undertook this habituation step twice daily with a ~40 min interval (while other mice were being run in the maze). Days 5 and 6 familiarized the mice to the combination of moving doors and food rewards; food rewards were present in all arms, the doors were operated as on days 1 and 2 and the trial was concluded after the mice had entered each arm at least once (only the first visit was rewarded). All mice in the cohort undertook two such sessions per day.

Habituation was followed by 9 days of RAM working memory training (Supplementary Videos 2-4). All arms were baited with diluted condensed milk (40 μl) at the start of each trial, which comprised 4 "forced runs" followed by a "free run". In the forced runs, mice were initially confined to the central hub for 5 s, then admitted sequentially to 4 pseudo-randomly selected arms, no more than 3 of which were adjacent arms. Mice were confined to each arm for 15 s while consuming the reward, before opening the door to that and the subsequent arm simultaneously. Following their fourth and final forced run, they were confined to the central hub for 5 s before all doors opened together (and stayed open for the remainder of the task), allowing them to visit the 4 as-yet-unvisited arms and collect the remaining rewards. Once all rewards were collected, mice were allowed to enter 4 additional arms before being returned to their home cage, reinforcing the fact that rewards are not replenished. Each mouse completed 6 trials per day with a between-trials interval of ~45 min (while other mice were being tested). Working memory errors were recorded if mice entered an arm they had previously visited during either the forced or free runs. Trials were scored by "success rate" [4/ (4 + working memory errors)]. For example, if 4 arm visits were required to recover the 4 remaining rewards, the success rate was 4/4 or 100% (a "perfect trial"). If 8 arm visits were required, the success rate was 4/8 or 50%. Mice that scored ≥10 perfect trials over the 9 days of RAM training were defined as "good performers", mice scoring ≤5 perfect trials were "poor performers".

In the early stages of the test, mice tended to visit arms sequentially in either a clockwise or anticlockwise direction (daisy-chaining) (Supplementary Videos 2-4). We can calculate the average score that it is possible to attain by this strategy. For example, for the forced-arm pattern 1246 (numbered clockwise), daisy-chaining clockwise from arm 1 during the free run requires 8 arm visits to recover the remaining rewards (which are in arms 3578) (4 errors, 50% score); starting from arm 2 requires 7 visits (3 errors, 57% score); starting from arm 3 requires 6 visits (2 errors, 67% score) etc., average over all 8 start-arms ~55%. Performing the same calculation for all the forced-run arm patterns that we employed gives an overall average score of ~55%. Control mice scored >75% after training, so they did not rely on daisy-chaining.

## Novel object recognition task (NOR) and object location task (OLT)

Mice were allowed to explore an acrylic 30 cm × 30 cm × 40 cm white Perspex open field box for 10 min (habituation stage). External visual cues above the open field box included a wall-mounted book shelf, ceiling lights and CCD camera to record their performances. Two identical objects (non-toxic plaster models ~10 cm high) were then placed in the box in a symmetrical arrangement and the mice were allowed 5 min to investigate and become familiar with the objects. Either 10 min or 24 h later one of the objects was replaced with a novel object of similar size and material that the mice had never experienced before. At the same time the familiar object was replaced with an identical version that had been spray-cleaned with 70% ethanol and dried thoroughly at least 10 min previously. After object replacement the mice were allowed to investigate for another 5 min. The whole procedure including the exploratory stage was recorded using a video camera mounted on the ceiling. The time mice spent interacting with the objects – defined as physical contact with the object, or investigative activity in which the nose is pointing towards the object and no more than 2 cm from it – was assessed from the videos by two independent observers, blind to genotype. The NOR discrimination index is $(t_n - t_f)/(t_n + t_f)$, where $t_n$ is time spent interacting with the novel object and $t_f$ is time interacting with the familiar object[87], averaged between the two observers. A prior control experiment confirmed that mice did not spend more time interacting with one of the objects over the other when they were both introduced into the open field box at the same time.

OLT was carried out as for NOR, with one modification. During the final test phase, instead of replacing one of the two identical objects with a novel object, one of them was moved to a new location in the box. The position of the displaced object was varied among mice. The OLT discrimination index is $(t_n - t_f)/(t_n + t_f)$, where $t_n$ is time spent interacting with the object in the new location and $t_f$ is time interacting with the object in the familiar location.

## Open field test (OFT)

The initial habituation stage of the NOR and OLT tasks, in the absence of any objects, doubled as an open-field test. Ten each of the NOR and OLT trials were selected randomly for analysis, prior to the tests having been conducted. ActualTrack software (Actual Analytics Ltd, Edinburgh, UK) was used to track mouse movements in during the 10 min test. The mean speeds and distances travelled by mice were calculated by the software, along with heat maps and bird nest maps of their trajectories.

## Histology and cell counts

Following behavioural tests, mice were transcardially perfused with 4% (w/v) paraformaldehyde (PFA, Sigma) in phosphate-buffered saline (PBS). Brains were dissected and post-fixed by immersion in 4% PFA overnight at 4 °C. The following day, brain tissue was cryoprotected in 20% (w/v) sucrose in diethyl pyrocarbonate (DEPC)-treated PBS until the tissue sank, before embedding in OCT compound

(Tissue-Tek) for cryo-sectioning. Coronal brain cryosections (25 μm) were collected and stored in 0.02% (w/v) sodium azide in PBS at 4 °C until needed. Immunostaining was as previously described[10,11]. To permeabilize the tissue and block nonspecific binding, sections were treated with blocking solution [10% (v/v) fetal bovine serum (FBS), 0.5% (v/v) Triton X-100 in PBS] for 2 h at 20-25 °C. Primary and secondary antibodies were diluted in 0.1% (v/v) Triton X-100 and 5% FBS in PBS. Primary antibody incubation was overnight at 4 °C. Secondary antibodies were Alexa-variants (Invitrogen): anti-chicken 488 nm (goat, Invitrogen, A11039, 1:10,000), antirabbit 488 nm (donkey, Invitrogen, A21206, 1:1000), antirabbit 568 nm (donkey, Invitrogen, A10042, 1:1,000), antirabbit 647 nm (donkey, Invitrogen, A31573, 1:1000), anti-mouse 568 nm (donkey, Invitrogen, A10037, 1:1000), anti-mouse 647 nm (donkey, Invitrogen, A31571, 1:500). These were applied together with Hoechst 33258 DNA dye (Sigma-Aldrich, 0.2 μg/ml) for 2 h at 20–25 °C. Primary antibodies were anti-Olig2 (rabbit, Merck AB9610, 1:500), monoclonal anti-Adenomatous polyposis coli (APC) (clone CC1, mouse, Calbiochem OP80, 1:200), anti-YFP (chicken, Aves labs, 1:1000), anti-Pdgfra (rabbit, Cell Signalling Technology 3164 S, 1:200), anti-Caspr clone K65/35 (mouse, Merck MABN69, 1:300), anti-Na$_V$1.6 (rabbit, Alomone ASC-009,1:500), anti-c-Fos (rabbit, Abcam ab190289, 1:1000). EdU detection using the Alexa Fluor 555 Click-iT kit (Invitrogen) was performed prior to blocking, according the manufacturer's instructions. For node analysis, anti-Na$_V$1.6 and anti-Caspr were applied together for 3 days at 4 °C, and secondary antibodies overnight at 4 °C[88]. For c-Fos immunolabeling, coronal brain sections were treated with a blocking solution containing 10% (v/v) FBS, 0.1% (v/v) Triton X-100 in PBS for 1 h at 20–25 °C. Primary anti-c-Fos, diluted in blocking solution, was added overnight at 4 °C. Secondary antibody (1:500) was applied for 1 h at 20–25 °C followed by Hoechst 33258 as above. Antibody lot numbers are given in the accompanying Reporting Summary.

Confocal images were taken using a Zeiss 880 Airyscan at 0.8-1.2 μm Z-spacing using Zen Black software and processed for analysis using Zen Blue. Cells were counted in tiled coronal images (20 μm Z-stacks) from the corpus callosum (Bregma +1.0 mm), anterior cingulate cortex (Bregma +1.0 mm), prelimbic and infralimbic cortex (Bregma +1.4 mm), hippocampus, fimbria and mediodorsal thalamus (Bregma −1.8 mm) − two to four sections from each region per brain, from three or more brains of each experimental group. Images were taken by S.G.N., re-labelled by M.G. before counting by S.G.N., blind to genotype, then decoded by M.G. Prism 9.0 (GraphPad) was used for statistical analysis. Nodes of Ranvier were counted in 5 μm thick Z-stacks in coronal sections of corpus callosum (Bregma +1.0 mm).

## c-Fos quantification

c-Fos fluorescence intensity was assessed as described previously[5]. Images of the ACC were collected from 3 sections per mouse, using uniform acquisition parameters to allow fluorescence intensity comparisons between groups. c-Fos$^+$ cell nuclei were outlined with the Freehand ROI tool in IMAGEJ, and the area, integrated density and mean grey value of the nucleus were measured. Fluorescence intensity corrected for background fluorescence was calculated as integrated density minus (area of selected cell x mean fluorescence of background) and expressed as arbitrary units (AU). c-Fos$^+$ cells were counted automatically using IMAGEJ. Cell signal was segmented from the background by thresholding and positive cells counted using the Analyse Particles tool with a 10 μm$^2$ minimum size cut-off. RAM training was performed by SN and coded tissue samples passed to MS for c-Fos quantification, then decoded and plotted by SN.

## Node and paranode length measurements

Node and paranode lengths were analyzed as described by Arancibia-Cárcamo et al.[89]. Briefly, confocal images of the anterior corpus callosum (Bregma +1.0 mm) were taken at 0.38 μm optical slice intervals.

Using ImageJ, images were background subtracted and maximum intensity projections were generated using a maximum stack thickness of 2.32 μm. A line intensity profile was drawn in IMAGEJ, spanning both Caspr-immunolabelled paranodes flanking nodal Na$_V$1.6 immunolabelling. The length of the node was then calculated using a MATLAB (The MathWorks Inc.) script (provided by Tania Quintela-López and David Attwell, UCL) that estimates the distance between the half-maximum intensities of Caspr immunofluorescence at each end of the node. To assess paranodal length, we modified the script to estimate the distance between the half-maximum Caspr immunofluorescence intensities at either end of the paranode. Total length of the node/paranode structure was the sum of the lengths of the node and both paranodes.

## Electron microscopy

One day after RAM training, mice were perfusion-fixed with 2.5% (v/v) glutaraldehyde and 2% (w/v) PFA in 0.1 M sodium cacodylate buffer (pH 7.4). Brains were post-fixed by immersion in the same fixative overnight at 4 °C, before transferring to 0.1 M PBS for shipment from UK at ambient temperature. In Japan, thick sagittal slices were prepared at <1 mm thickness from 4.5 × 2 mm tissue blocks containing the whole length of the corpus callosum. The sections were immersed in 1% (w/v) osmium tetroxide solution for 2 h at 4 °C, dehydrated through a series of graded alcohols and embedded in Epon 812 resin (TAAB Laboratories, UK). Ultrathin parasagittal sections (100 nm) were cut on an ultramicrotome (Ultracut UCT, Leica) and collected on tin-coated glass slides, stained with uranyl acetate and lead citrate and imaged in a scanning EM equipped with a back-scattered electron beam detector (Hitachi SU8010) at 1.0–1.5 kV accelerating voltage, for quantifying axon-myelin units and nodes of Ranvier. Cross-sectional profiles of myelinated axons, paranodes and nodes (Fig. 5) were counted in Japan by K.T., blind to genotype; brain samples were codified prior to shipping and the data subsequently decoded in the U.K.

## Statistics and reproducibility

Statistical significance was determined using GraphPad Prism (GraphPad Software, CA, USA) and OriginPro software. Repeated measures two-way Analysis of Variance (ANOVA) was used to compare behavioural performance across groups and time in the T-maze and RAM tasks (Fig. 1). Unpaired Student's t-tests were used for other behavioural experiments (Supplementary Fig. S1), $p$ values were corrected for multiple comparisons using Šídák or Tukey post-tests as specified. Data were presented as mean ± s.e.m. Normality of data distribution was assessed using Kolmogorov−Smirnov and Shapiro-Wilk tests. Where normality criteria were not satisfied (e.g. some of the cell counts of Fig. 3 and Fig. S2), data were compared using the Kruskal-Wallis nonparametric test, $p$ values were corrected for multiple comparisons using the Benjamini-Krieger-Yekutieli (BKY) false discovery rate test[88] (Supplementary Data 1, 2) and data were presented as median ± interquartile range (25%-75%). A simple linear regression model using a least squares regression without weighting was applied to good- and poor-performers in Fig. 4 to generate a line of best fit with 95% confidence bands. The goodness-of-fit of the line is represented by the coefficient of determination R$^2$. Algebraic equations for the lines of best fit together with their R$^2$ values are given in Supplementary Data 3, 4). Each line of best fit was subjected to an extra sum-of-squares F-test against a theoretical gradient of zero to determine whether the gradient of the line was significantly different from zero, demonstrating positive or negative correlation.

All immunolabelling experiments were repeated at least 3 times, with comparable results, and representative images chosen for display.

## Display items

Illustrations for figures were created using BioRender and Adobe Photoshop Elements 2020. Graphs were generated in GraphPad, annotated and arranged in Adobe Photoshop Elements.

**Reporting summary**

Further information on research design is available in the Nature Portfolio Reporting Summary linked to this article.

## Data availability

Source data are provided with this paper.

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

## Acknowledgements

We thank our colleagues at UCL for help, advice and discussion. We especially thank Tomohito Hanasaka and all members of the Technical Support Center for Life Science Research at Iwate Medical University for their expertise in electron microscopy and morphometric analysis. We also thank Tania Quintela-López and David Attwell (UCL) for advice on node/paranode immunolabelling and analysis and for providing a MATLAB script. This work was supported by the Wellcome Trust (108726/Z/15/Z and 214286/Z/18/Z to W.D.R.) and a Sanming Project (SZSM201911003) funded by the Municipal Government of Shenzhen, China to W.D.R, H.L. and others. H.J.-B. and the Wellcome Centre for Integrative Neuroimaging were supported by the Wellcome Trust (222446/Z/21/Z and 203139/Z/16/Z).

## Author contributions

T.S. and W.D.R. conceived the project. W.D.R. and H.L. obtained funding. T.S., S.N., M.S., M.K., K.T. and W.D.R. designed the experiments with input from H.L., C.S.B., H.J.-B. and D.B. Experiments were performed by S.N., T.S., M.S., Y.J., K.T. K.O. and M.K., with technical assistance from M.G. K.O. provided EM facilities and expertise. S.N., T.S., M.S., and W.D.R. interpreted the data and wrote the paper with input from all authors.

## Competing interests

The authors declare no competing interests.
