## [Peer Review File · Nature Communications]

Oligodendrocyte dynamics dictate cognitive performance outcomes of working memory training in miceREVIEWER COMMENTS

Reviewer #1 (Remarks to the Author):

In the manuscript, Shimizu and Nayar et al. describe the importance of OPC/OL generation and myelin formation for working memory. They employed T-maze and 8-arm radial maze combined with 'win-shift' protocol, to train and assess the working memory of the mice. After 9 days of RAM training, the control animals could perform much better than their mutant littermates, the MyRF OPC-cKO mice. In addition, in the control animals, the performance of the animals were positively correlated with the number of newly generated OPCs in many working-memory related brain regions, including corpus callosum, anterior cingulate cortex, prefrontal cortex, hippocampal CA1 region and fimbria, already at one day after training. Further immunohistological and ultrastructural analysis of myelin showed that the nodal density was increased while the internodal length might have been decreased in the good performers compared to home-caged animals. In general, how OL lineage cells contribute to working memory is intriguing topic, although new OL and myelin generation in working memory is somewhat expected from the established work in different paradigms (Xin et al., Nat Rev Neurosci, 2020) as well as in working memory (Nguyen et al., Eneuro, 2020). The story would be more attractive if the authors address the mechanism why and how working memory requires the OPC and OL generation. For instance, whether and how neuronal activities are triggered during the establishment of working memories? If yes, how do they transmit the signals to the OL lineage cells? The lack of further mechanistic or pathway relevant insights into regulation of OL generation required by working memory remains an issue making the study more descriptive.

Other issues that require further experiments and/or careful discussion are detailed below:

In the mutant mice, is OPC generation also altered/suppressed? or only OLs can not be generated? How about neuronal activities or neural circuit? Are these changed in mutant mice compared to the controls, and in the good performers compared to the poor ones? Because the change in the working memory can be translated from neural circuit change.

The authors could clearly show that the OPC and OL generation is positively correlated with the performance of the control animals. But it is still largely unclear whether this is because the poor performers cannot generate OL thereby they cannot perform better, or because they perform poor (by other unknown reasons) so they generate less OL? The causal link is not very clear. In addition, the MyRF cKO mice cannot (or in a very low extent) generate new OLs anyhow, which might impair their long-term memory (Xin et al., Nat Rev Neurosci, 2020). Indeed, as the authors showed in the figure 1C, D and supplementary figure 1J, mutant mice performed as 'good' as the control animals at the first four days. Only from the 5th day, they start to exhibit cognitive deficit. Therefore, the long-term cognition of the mutant mice should also be assessed.

Authors analyzed newly generated OPC and OL, and myelination. Did they also compare the total density of OL in the regions? If the density remains unchanged, do the pre-existing OLs get eliminated and get replaced by the newly formed OLs? Also, increased number of internodes are from newly generated OLs or pre-existing OLs also form more myelin sheaths? Authors could utilize PDGFR α -CreERT2 x reporter mice (without MyRF flox), inject TAM right before the RAM test, and

compare the newly formed myelin (reporter+) between the good and poor performers of the control animals. It will only sparsely label the OPCs, but still can distinguish newly formed OLs from pre-existing ones.

In addition, authors observed increased number of nodal/paranodal density in the corpus callosum of good performers compared to home-cage animals. Did they compare the good performers with poor ones? Authors stated in the text (page 11) that there was no difference in the numbers of new OLs between poor performers and home-caged controls. However, the proper comparison should be done between the good and poor performers, to support their hypothesis that individual performance of working memory is dependent on OL dynamics. In addition, to convincingly address the functional improvement of myelination, authors could also compare g-ratio (as they already have EM images) and the compound action potential conduction velocity in CC.

How about myelin changes in other brain regions, PFC, hippocampus, ACC and Fimbria between good and poor performers?

Minor points

When were the NOR and NOL tasks performed? Also after 3 weeks of TAM injection? did the OL density already change before any of the behavior test?

In figure 2E-H, the arrows are in the same color as the immunostaining, it is hard to recognize the arrows especially in green channels.

In supplementary figure D and E, authors wrote OLT (object location task) but in the figure legend is NOL (new object location).

Throughout the manuscript, it is not clear how the data are presented in the dot plots, Mean \pm SEM (Figure 1) or Median with 25-75% interquartile (in suppl. Fig.2)?

Did the control animals also got Tamoxifen injection?

In Figure 5, it is not clear when the samples were collected for the EM analysis, i.e. at how many days after RAM?

Immunostainings can be shown together with the quantifications, as there is quite some space left in Figure 3 and Supplementary Figure 2.

Reviewer #2 (Remarks to the Author):

This paper by Shimizu and colleagues explores the role of oligodendrogenesis in working memory, and adds to an emerging literature on the importance of adaptive myelin plasticity in cognition. The main findings are as follows. Using conditional Myrf mice, they find that the generation of new oligodendrocytes is necessary for working memory using T-maze and radial arm mazes (RAM) (but not for simple discrimination learning or recognition memory). They find that learning in RAM stimulates the generation of new oligodendrocytes in both the corpus callosum and medial prefrontal cortical regions, and that levels correlate well with performance. Finally, they show that RAM learning induces myelination in the anterior corpus callosum, consistent with the idea that oligodendrocyte generation and de novo myelination contribute to cognitive processes.

This is a straightforward paper, with findings that are largely consistent with two recent papers using water maze and contextual fear conditioning assays (Pan et al 2020, Steadman et al 2020).

Particularly notable here is the careful behavioural analysis, together with the thoughtful presentation and interpretation of the behavioural data. These analyses exclude several alternate 'non-mnemonic' interpretations of the results (e.g., motor performance confounds).

My comments and suggestions are all minor.

1. The authors suggest a distinction between the current analyses and those in the Pan et al and Steadman et al papers that looked at consolidation, rather than initial learning (see, for example, paragraph 3, introduction and first paragraph of discussion). For the Steadman paper this is not the case. In that study, the authors also looked at the effects of conditional Myrf deletion prior to training on initial learning in the water maze, and deficits were apparent in a probe test 1 day after the completion of training (Figure 2). This detracts a little from the novelty of the current work.
2. Elevated OLPs at day 14 (Fig 3C, F). might this suggest this population is contributing to some form of consolidation? (Similar to Steadman/Pan)?
3. Figure 3 caption. Please define H, G, P and M (or spell out).
4. Please add labels to y-axes on all figures.
5. Last paragraph of results should be moved to discussion.

Reviewer #3 (Remarks to the Author):

In this study, Shimizu et al. attempt to demonstrate that the production of oligodendrocyte precursor cells (OPCs) and new myelin is required to improve performance in spatial working memory tasks. The authors show that oligodendrogenesis is necessary to improve task performance in a radial arm maze (RAM) and T-maze by conditionally knocking out myrf in OPCs, preventing new myelination from occurring. These cKO mice were unable to improve their performance after day 6 in the T-maze task and day 5 in the RAM. The authors also quantified the number of EdU+/PDGFRa+ cells and EdU+/CC1+ cells to determine if there was a correlation between task performance and the amount of oligodendrogenesis. Mice that performed well in the RAM show increased proliferation of OPCs and differentiation into mature oligodendrocytes (OLs) in the corpus callosum (CC), anterior cingulate cortex (ACC), PLC/ILC, and fimbria 1 and 14 days post-RAM training. Intriguingly, they found that at the individual level, the amount of EdU+/PDGFRa+ cells found in the CC, ACC, PLC/ILC, CA1, and fimbria 1-day post-RAM positively correlated with performance in the RAM. The same was

not true for Edu+/CC1+ cells in all regions, except in the CC ($R^2 = 0.61$). Finally, the authors immunolabeled Nav1.6 channels, which are present in the axonal membrane at nodes, and the adhesion molecule Caspr to quantify changes in myelination in the CC. Through EM of CC sections, they found that there was an increase in nodes and paranodes in RAM-trained mice, but no significant difference in the number of myelinated axons.

From these experiments, the authors concluded that increased OPC proliferation and differentiation are necessary to increase successful performance during spatial working memory tasks. They suggest that new myelination of PFC axons may improve working memory by reducing the energy needed to propagate action potentials and provide newly myelinated segments of axons with metabolic substrates required to facilitate energy production. Additionally, new myelination of distinct hippocampal and PFC neuronal assemblies could play a role in modulating synchrony between these regions, subsequently impacting performance during working memory tasks. While the authors convincingly demonstrate that OPC proliferation and differentiation are necessary to improve performance in spatial working memory tasks, they do not rigorously demonstrate quantifiable changes in myelination as a result of learning these tasks. As is, the manuscript would require further experiments to show that performance in a T-maze and RAM is causally related to newly myelinated segments of axons.

Major Points:

- The main concern is that there is no quantifiable evidence of new myelination, despite implying that the increase in differentiated OLs results in an increase in myelination. It is not necessary that all cells that differentiate go on to myelinate new segments of axons. In Fig1 and supplementary Fig 2, the authors show that in good performers there is a significant increase of Edu+/CC1+ cells 1 and 14 days post-RAM compared to controls and poor performers, but what about the number of myelinating oligodendrocytes? What is the justification for no significant difference in either Edu+/PDGFRa+ or Edu+/CC1+ cells between good performers and controls in Fig 1H, 1J, and supp. Fig 2D?
- In Fig. 5, while immunostaining for Nav1.6 and Caspr they found no significant differences in myelinated axon profiles between controls and RAM-trained mice, but they did find differences in the number of node/paranode profiles using EM in the CC. Is this consistent with the other regions that show an increase in CC1+ cells 14 days post-RAM? Why does the immunostaining data not align with changes seen using EM?
- In the manuscript (page 13, paragraph 3) the authors state that the mediodorsal thalamus is an important region implicated in working memory, supported by the literature (Bolkan et al., 2017), but do not show any data related to this brain region. Were there no changes in OPC proliferation/differentiation here, if so why not?
- The title does not accurately reflect the main findings of the paper. The authors do show that there is a correlation between performance and newly proliferating OPCs in one experiment, but it seems that the main findings are that oligodendrogenesis is necessary to improve working memory performance.
- For all experiments, only male mice were used. Why were female mice not used? Is there sufficient evidence in the literature that the OPC dynamics and performance in spatial working memory tasks are the same between the sexes?

Minor Points:

- Missing citations on page 12 line 6,

- In the last sentence of the summary, "...indicating a key role for adaptive OL genesis and myelination in cognitive processing" the conclusion is too vague. The authors do not show enough evidence that new myelination is occurring and implying this process is necessary for all cognitive processing is not an accurate conclusion from their data.
- On page 2 paragraph 3, the authors state that there is preferential myelination of electrically active axons. However, the implication is that "inactive" axons are generally unmyelinated which is not necessarily true given the probability of an oligodendrocyte myelinating an axon is contingent on a variety of criteria. The study in reference was also achieved through pharmacological manipulation which may not be consistent with how myelination occurs naturally in-vivo.
- On page 11 line 16, the authors state that the maze training results in EdU+/CC1+ OLs forming new myelin sheaths but do not quantifiably show this.

RESPONSE to REVIEWERS

We thank the reviewers for their time and attention to detail in reviewing our original submission, and for their helpful comments. We have taken their suggestions and criticisms to heart and have carried out two new sets of experiments that are presented in two new Figures 6 and 7, with additional descriptions and discussion in the text. We copy below the comments of the three reviewers verbatim (*in italics*) followed by our responses. All changes to the text of the m/s are highlighted in our resubmitted paper.

Reviewer 1

We thank this reviewer for his/her detailed reading of our m/s and for his/her many insightful comments. Going through his/her review point- by-point:

“In general, how OL lineage cells contribute to working memory is intriguing topic, although new OL and myelin generation in working memory is somewhat expected from the established work in different paradigms (Xin et al., Nat Rev Neurosci, 2020) as well as in working memory (Nguyen et al., Eneuro, 2020).”

We are pleased that this reviewer finds the general topic of our study intriguing. We agree that the work of Pan et al. 2020 and Steadman et al. 2020 on consolidation of fear and spatial memory, together with our own earlier work on motor learning (McKenzie et al., 2014; Xiao et al., 2016) and related work from other labs, come together to support the notion that adaptive myelination could be a key factor in many cognitive processes, as reviewed e.g. by Xin et al. 2020. However, we do not think it is fair to say that a role for OL generation specifically in working memory is “somewhat expected”. We certainly do not think the striking proportionality we have demonstrated between working memory performance and OL dynamics could have been expected.

“The story would be more attractive if the authors address the mechanism why and how working memory requires the OPC and OL generation. For instance, whether and how neuronal activities are triggered during the establishment of working memories? If yes, how do they transmit the signals to the OL lineage cells? The lack of further mechanistic or pathway relevant insights into regulation of OL generation required by working memory remains an issue making the study more descriptive.”

How adaptive myelination interacts with the neuronal circuitry to enhance cognitive processes, including working memory training, is certainly one of the holy grails of research in this area. We have chosen to focus on one side of the problem, because it is important to establish the fundamentals before we can hope to know what larger questions to ask.

To start to ask how neuronal activity might be affected in good-performers versus home-cage or poor-performing controls, we have now performed c-Fos immunolabelling of the prefrontal cortex in each of these conditions. This admittedly is an indirect measure of neuronal activity, at best, but is a reasonable place to start. At this level of analysis we found no significant differences in the numbers or average fluorescence intensity of c-Fos-

expressing cells in good- versus poor-performers or home cage controls in the mPFC. These data are presented in a new Figure 7 and described in the text (highlighted on p4 and p15 with a short discussion on p17). Our conclusion is that there is a more obvious positive correlation between working memory performance and OL dynamics than there is with global neuronal activity estimated by c-Fos expression. Intriguingly, there appears to be a weak *negative* correlation between c-Fos⁺ neuron numbers and performance (Fig. 7F). This will be fascinating to follow up in the future, although it falls outside the scope of the present paper. The lack of a direct relationship between bulk neuronal activity in the mPFC and OL dynamics indicates that more nuanced physiological investigations are required to determine how neuronal circuit activity is affected by RAM training, while at the same time highlighting the prominent role played by OLs and myelin. We briefly discuss this on p17:

The dynamic changes to OL lineage cells and myelin that we observe are very likely triggered directly or indirectly by changes in circuit activity as the animals learn. The newly-generated OL lineage cells must in turn modify the behaviour and activity of the circuits with which they interact, so that learning and memory formation is a reciprocal dialogue between neurons and OL lineage cells – potentially even extending to other cells such as astrocytes, microglia and perivascular cells. We attempted to detect differences in the level of activity of ACC neurons in good- versus poor-performers or home cage controls, by immunolabelling for the immediate-early gene c-Fos. Intriguingly – and against expectation – there was a trend towards a lower number and intensity of c-Fos⁺ neurons in good-performers versus controls, and a tighter clustering around the mean, although this did not reach statistical significance (Kruskal-Wallis non-parametric analysis). There was even a weak negative correlation between the number of c-Fos⁺ neurons and working memory performance. If these observations can be substantiated by future studies, it invites speculation that one outcome of altered OL dynamics might be to limit activity to a reduced subset of task-relevant neurons, potentially cutting out background noise and allowing more channeled and efficient problem-solving behaviour in good- versus poor-performers.

“In the mutant mice, is OPC generation also altered/suppressed? or only OLs can not be generated? How about neuronal activities or neural circuit? Are these changed in mutant mice compared to the controls, and in the good performers compared to the poor ones? Because the change in the working memory can be translated from neural circuit change.”

Myrf is known to be expressed in newly-differentiated and mature OLs but not in undifferentiated, cycling OLPs. Therefore *Myrf*-cKO should not affect OLP proliferation and this is what is found (e.g. Emery et al., Cell 2009; McKenzie et al., 2014). We have now included new data on EdU incorporation into OLPs (Fig. 3B,C), confirming similar rates of OLP division in *Myrf*-cKOs, home cage controls and poor-performers. It is just that OLPs cannot differentiate to form new OLs in the *Myrf*-cKO.

“The authors could clearly show that the OPC and OL generation is positively correlated with the performance of the control animals. But it is still largely unclear whether this is because the poor performers cannot generate OL thereby they cannot perform better, or because they perform poor (by other unknown reasons) so they generate less OL? The causal link is not very clear. In addition, the MyRF cKO mice cannot (or in a very low extent) generate new OLs anyhow, which might impair their long-term memory (Xin et al., Nat Rev Neurosci, 2020). Indeed, as the authors showed in the figure 1C, D and supplementary figure 1J,

mutant mice performed as 'good' as the control animals at the first four days. Only from the 5th day, they start to exhibit cognitive deficit. Therefore, the long-term cognition of the mutant mice should also be assessed."

The point of the behavioural experiments with *Myrf*-cKO mice was to show that de novo OL generation is required for mice to improve their performance during working memory training in the RAM. On this basis, we argue that the reason some mice perform poorly relative to others is because they fail to increase their output of OL lineage cells, not the other way round. However, as pointed out by the reviewer, this raises the interesting question of why do they fail to increase OL production? We have no experimental insights into this at present, but we think that poor-performers (and *Myrf*-cKOs) fail to make a strategic shift away from random arm visits or sequential "daisy-chaining", to a more efficient strategy based on working memory. We think that the reason all mice perform similarly up to day 4 of RAM training is because it takes that long for some of the mice (the good-performers) to make this strategic shift and hence move ahead of the pack. It is possible that this behavioural shift, and the successful outcome it generates, provides a rewarding stimulus that drives OLP proliferation and/or differentiation and that this, in turn, is required to capture or consolidate the circuitry underlying the shift. Further improvements are made on top of this, in a reiterative process that turns exploratory changes in circuit activity into more stable changes which then allow further refinement of circuit activity, which are again consolidated, and so on. We discussed the idea of strategy-switching in the Methods section of our original paper but have chosen not to develop this further in the revised paper because it is pure speculation at this point.

We agree that it would be very interesting and informative to look at the persistence (memory) of working memory training in the post-training period. If memory consolidation is deficient in the *Myrf*-cKOs, then we might expect their performance in the RAM to decrease rapidly to pre-training levels after training has ceased, whereas the performance of control mice should persist longer. We have carried out a preliminary experiment that appears to support this version of events. After 14 days post-training the performance of *Myrf*-cKO mice had decreased to baseline while the controls were at ~80% of their maximum performance. However, to do these experiments justice would require additional time points post-training, each with larger numbers of mice than we have examined thus far, which would take us at least another year. We hope this reviewer will agree that this is a non-trivial task that should be the subject of a separate study and that lies outside the reasonable scope of our present paper.

"Authors analyzed newly generated OPC and OL, and myelination. Did they also compare the total density of OL in the regions? If the density remains unchanged, do the pre-existing OLs get eliminated and get replaced by the newly formed OLs? Also, increased number of internodes are from newly generated OLs or pre-existing OLs also form more myelin sheaths? Authors could utilize PDGFR α -CreERT2 x reporter mice (without MyRF flox), inject TAM right before the RAM test, and compare the newly formed myelin (reporter+) between the good and poor performers of the control animals. It will only sparsely label the OPCs, but still can distinguish newly formed OLs from pre-existing ones."

It is not feasible to compare the total density of OLs throughout all the regions examined, because of their abundance (~3000 OLs/100 μ m² in the corpus callosum, for example). The

newly-formed OLs are a small fraction of this pre-existing population and not easily detected against this background. However, we adopted this reviewer's suggestion of sparsely labelling newly-formed OLs using a fluorescent reporter together with *Pdgfra-CreER*^{T2}. We used *Tau-mGFP* as reporter, which reveals whole cell morphology including myelin sheaths. These new experiments, shown in our new Fig. 6, provide convincing evidence that the new OLs formed during RAM training indeed make new myelin sheaths. This was also one of the major points raised by reviewer 3 (please see below, pages 6/7).

In addition, we started to examine the functional maturity of the newly-formed myelin sheaths by immunolabelling nodes of Ranvier (Caspr and NaV1.6). This analysis revealed that, at the time of analysis (1-day post RAM training), a majority of the new myelin sheaths in gray matter are heminodes that are not associated with clustered NaV1.6. However, there are clearly also some mature nodes with two closely spaced paranodes separated by a cluster of NaV1.6. This preliminary analysis raises a new set of questions about the developmental sequence of node formation on the one hand and, on the other hand, the role of increased conduction speed in the temporal sequence of learning and memory formation. These are exciting questions to follow up in future but are outside the scope of our present paper. We discuss these findings on p16:

Experiments with the *Tau-mGFP* reporter demonstrated that many of the newly-formed OLs in cortical grey matter at 1-day post-training formed new myelin sheaths. However, only a fraction of the new sheaths in gray matter formed normal-appearing nodes of Ranvier with clustered Na_v1.6, the majority forming heminodes that did not cluster Na_v1.6. It is unlikely that these heminodes contribute to accelerated conduction velocity, but they and their associated internodes might provide other functional input to the axons of local interneurons and long-range output neurons. Most of the additional OLs that form in the ACC of good-performing animals are eliminated in the 14 days post-training, so their influence on the neuronal circuitry is presumably transient.

We also visualized many newly-formed (mGFP⁺) OLs in the sub-cortical white matter (CC) that were synthesizing myelin at 1-day post-training. The functional maturity of these sheaths was difficult to assess, but some of them terminated in normal-appearing nodes, while others terminated at heminodes, like their counterparts in gray matter. It is possible that heminodes normally mature into full nodes as their myelin sheaths elongate, gather Na_v channels and collide with other, pre-existing heminodes. Further experiments will be required to investigate this. The fact that heminodes are observed at all suggests that the sparse pattern of myelination observed on the axons of projection neurons in the cortex⁴⁴ might be maintained as those axons traverse the corpus callosum.

Most of the extra CC1⁺ OLs that formed in the CC during training survived for at least 14 days post-training (Fig. 3D, G). It will be interesting to discover whether all the myelin sheaths of these surviving OLs assemble mature, functional nodes and whether their survival lifetime correlates with the lifetime of training-induced working memory improvement.

"In addition, authors observed increased number of nodal/paranodal density in the corpus callosum of good performers compared to home-cage animals. Did they compare the good performers with poor ones? Authors stated in the text (page 11) that there was no difference

in the numbers of new OLs between poor performers and home-caged controls. However, the proper comparison should be done between the good and poor performers, to support their hypothesis that individual performance of working memory is dependent on OL dynamics. In addition, to convincingly address the functional improvement of myelination, authors could also compare g-ratio (as they already have EM images) and the compound action potential conduction velocity in CC.”

“How about myelin changes in other brain regions, PFC, hippocampus, ACC and Fimbria between good and poor performers?”

We thank the reviewer for these suggestions, which would certainly improve our m/s. However, we cannot comply because of technical and time limitations (the EM analysis in particular is extremely time- and labour-intensive) and, in any case, our new experiments with *Tau-mGFP* render the nodal analyses less critical to the overall argument that new OLs generate new myelin sheaths. Also, we do not currently have the capability to perform conduction velocity measurements in vivo, although we plan such experiments in the near future with collaborators in Japan. Altogether we do not think that our hypothesis, that individual performance of working memory is dependent on OL dynamics, is seriously in doubt and we hope that this reviewer agrees.

Minor points

“When were the NOR and NOL tasks performed? Also after 3 weeks of TAM injection? did the OL density already change before any of the behavior test?”

Yes, all behavioural tests were performed three weeks post-tamoxifen administration, to allow the mice to recover fully (at the doses we habitually use the mice lose weight temporarily and show other signs of discomfort). However, the *Tau-mGFP* reporter experiments of our new Fig. 6 were done using a smaller dose of tamoxifen and this allowed us to start the behavioural experiments earlier, within one week (allowing one week of dietary restriction).

“In supplementary figure D and E, authors wrote OLT (object location task) but in the figure legend is NOL (new object location).”

Thank you for pointing this out, it is now corrected.

“Throughout the manuscript, it is not clear how the data are presented in the dot plots, Mean \pm SEM (Figure 1) or Median with 25-75% interquartile (in suppl. Fig.2)?”

In Fig. 1 the data points are presented as mean \pm s.e.m (with correction for repeated measures but as median and 25-75% interquartile in Fig. 3 and Supplementary Fig. S2, as is now spelled out in the corresponding figure legends and in the Methods section.

“Did the control animals also got Tamoxifen injection?”

Yes, all controls received Tamoxifen. This is now stated in the Methods section, p19.

“In Figure 5, it is not clear when the samples were collected for the EM analysis, i.e. at how

many days after RAM?”

We now clarify in the main text (p11 and p12) that the tissue samples both for immunofluorescence and EM were collected 1-day post-training.

“Immunostainings can be shown together with the quantifications, as there is quite some space left in Figure 3 and Supplementary Figure 2.”

Thanks for this suggestion but this would require too much re-organization of the m/s at this stage, and would make for a large and unwieldy figure. We hope the reviewer understands.

Reviewer 2

“This is a straightforward paper, with findings that are largely consistent with two recent papers using water maze and contextual fear conditioning assays (Pan et al 2020, Steadman et al 2020). Particularly notable here is the careful behavioural analysis, together with the thoughtful presentation and interpretation of the behavioural data. These analyses exclude several alternate ‘non-mnemonic’ interpretations of the results (e.g., motor performance confounds). My comments and suggestions are all minor.”

We are grateful to this reviewer for his/her positive comments.

“1. The authors suggest a distinction between the current analyses and those in the Pan et al and Steadman et al papers that looked at consolidation, rather than initial learning (see, for example, paragraph 3, introduction and first paragraph of discussion). For the Steadman paper this is not the case. In that study, the authors also looked at the effects of conditional Myrf deletion prior to training on initial learning in the water maze, and deficits were apparent in a probe test 1 day after the completion of training (Figure 2). This detracts a little from the novelty of the current work.”

We thank the reviewer for pointing out that Steadman et al. found a deficit in short-term recall of learned spatial information. Steadman et al. went to some length to emphasize memory over learning, using the term “memory consolidation” once in the title of their paper and four times in the abstract, without mentioning “learning”. Their Fig 2G shows that the time taken for *Myrf*-cKOs find the water maze platform was the same as control mice on each of days 1-4 of training, despite their searching less systematically and spending less time in the platform zone (Fig. 2H). It is not clear whether the deficit in 1-day recall (“probe”, Fig. 2I) represents a deficiency in learning per se, or a deficiency of memory over the subsequent 24 hours. Therefore, we do not accept that Fig. 2 of Steadman et al. detracts significantly from the novelty of our own findings, which are concerned with a different cognitive task and which demonstrate a clear deficit in learning (skill acquisition) in *Myrf*-cKO mice.

“2. Elevated OLPs at day 14 (Fig 3C, F). might this suggest this population is contributing to some form of consolidation? (Similar to Steadman/Pan)? ”

The reviewer’s suggestion that OLPs might contribute to consolidation of working memory training is an interesting idea and certainly possible. In the original version of our paper we already entertained the idea that OLPs might play a direct role in promoting cognitive

performance, apart from their indirect role as a source of new OLs (Discussion, p13):

“... improvement of working memory might depend on direct OL differentiation, while OLP division and secondary OL differentiation consolidate the improvement. Alternatively, or in addition, enhanced OLP proliferation and the consequent increase in OLP population density, which persisted for at least 2 weeks post-training in the white matter, might itself lead to improved cognitive performance. There are reports that immature OL lineage cells in zebrafish and mice can exert effects on neurons that are unrelated to myelination, including synaptic pruning by OLPs.”

“3. Figure 3 caption. Please define H, G, P and M (or spell out).”

H,G,P and M were and are defined in a visual key within the figure itself (H=home cage, G=good performer, P=poor performer, M=Myrf-cKO). However, this is easily missed so we have now also defined the abbreviations in the relevant figure legends.

“4. Please add labels to y-axes on all figures.”

We have done this.

“5. Last paragraph of results should be moved to discussion.”

We have done this.

Reviewer 3

We thank this reviewer for his/her detailed reading of the m/s and his/her reasonable criticisms and suggestions. We deal with his/her points one-by-one below:

“While the authors convincingly demonstrate that OPC proliferation and differentiation are necessary to improve performance in spatial working memory tasks, they do not rigorously demonstrate quantifiable changes in myelination as a result of learning these tasks. As is, the manuscript would require further experiments to show that performance in a T-maze and RAM is causally related to newly myelinated segments of axons.”

“The main concern is that there is no quantifiable evidence of new myelination, despite implying that the increase in differentiated OLs results in an increase in myelination. It is not necessary that all cells that differentiate go on to myelinate new segments of axons. In Fig1 and supplementary Fig 2, the authors show that in good performers there is a significant increase of EdU+/CC1+ cells 1 and 14 days post-RAM compared to controls and poor performers, but what about the number of myelinating oligodendrocytes?”

This is also one of the major criticisms made by Reviewer 1; as explained in our response to that reviewer we have now conducted further experiments with the *Tau-mGFP* reporter line to address this issue explicitly, and we believe that we have established beyond reasonable doubt that increased numbers of new myelinating OLs are generated in good-performers in the RAM, compared to home cage controls. At least some of the new myelin sheaths made in RAM-trained mice were functional, judging by the presence of normal-appearing nodal structures. We acknowledge that the ideal comparison would have been between good- and

poor-performers but since we did not obtain any poor performers in this new experimental cohort, and since there was no significant difference between the number of EdU⁺CC1⁺ OLs generated in poor-performers and home cage controls in the experiments of Fig. 3, we hope that this reviewer will accept that we have made our best efforts to address this important point and that he/she is now convinced, as we are, that extra new myelin-forming OLs are indeed formed in response to successful working memory training.

“What is the justification for no significant difference in either Edu+/PDGFRa+ or EdU+/CC1+ cells between good performers and controls in Fig 1H, 1J, and supp. Fig 2D?”

We presume the reviewer was referring to Fig. 3 and Supplementary Fig. S2. We used non-parametric statistics (Kruskal-Wallis non-parametric test; p-values corrected for multiple comparisons using the Benjamini-Krieger-Yekutieli (BKY) false discovery rate test) for these data because some of them (the good-performers) were not normally distributed. These statistical tests are notoriously stringent and conservative, with the result that some comparisons that look very different (e.g. those picked out by the reviewer) do not pass the arbitrary significance threshold ($p < 0.05$) even though they look and undoubtedly are reproducibly different. Indeed, not all commentators agree that multiple comparison corrections are necessary or justifiable; without this correction the data referred to would easily reach the threshold of significance.

“In Fig. 5, while immunostaining for Nav1.6 and Caspr they found no significant differences in myelinated axon profiles between controls and RAM-trained mice, but they did find differences in the number of node/paranode profiles using EM in the CC. Is this consistent with the other regions that show an increase in CC1+ cells 14 days post-RAM? Why does the immunostaining data not align with changes seen using EM?”

We think that the immunostaining data do align with the EM data, the only difference being that the EM data pass the (arbitrary) statistical threshold of $p < 0.05$ while the immunolabelling data do not. The immunolabelling data (Fig. 5A,B) (nodes+paranodes per 100 μm^2) reach $p = 0.066$ while the EM data (Fig. 5I) reach $p = 0.016$, in the same direction. There is no misalignment here and these data are now strongly supported and to some extent superseded by our new data using the *Tau-mGFP* reporter (new Figure 6). All of these analyses are very labour-intensive, especially the EM analysis, so we focused on the region with the largest difference in new OL generation between good- and poor-performers (corpus callosum).

“In the manuscript (page 13, paragraph 3) the authors state that the mediodorsal thalamus is an important region implicated in working memory, supported by the literature (Bolkan et al., 2017), but do not show any data related to this brain region. Were there no changes in OPC proliferation/differentiation here, if so why not?”

We did not analyze the mediodorsal thalamus (MDT) in our original submission for no particular reason other than workload. However, in response to this reviewer's question we have now analyzed the MDT and include the new data in Supplementary Figs. S2 and S3. OL generation in the MDT in good- versus poor-performers was similar to the PLC/ILC, and there was a linear correlation between performance and OLP proliferation in the MDT ($R^2 = 0.67$) also similar to the PLC/ILC.

“The title does not accurately reflect the main findings of the paper. The authors do show that there is a correlation between performance and newly proliferating OPCs in one experiment, but it seems that the main findings are that oligodendrogenesis is necessary to improve working memory performance.”

The *Myrf*-cKO data do suggest that OL genesis is critical for performance improvement during working memory training in the T-maze and RAM. However, we also found that OLP division (EdU incorporation) correlated even more closely with performance (in phenotypically wild type mice) than new OL generation, raising the possibility that OLPs themselves could play a role in some aspect of training-induced learning (Figs 3,4, Supplementary Figs S2,S3). Ultimately, new OL generation depends on OLP division (or the OLP population would crash) so “OL dynamics” does, we believe, accurately reflect the findings of our study.

“For all experiments, only male mice were used. Why were female mice not used? Is there sufficient evidence in the literature that the OPC dynamics and performance in spatial working memory tasks are the same between the sexes?”

Until recently it has been customary to use male mice for behavioural studies, to avoid potential or perceived individual variation due to the female reproductive cycle – directly in females and indirectly in males through exposure to females at different stages of estrus. In our previous study (McKenzie et al., 2014) we used male mice for motor learning experiments, but also compared male and female mice and found no significant difference in motor ability. In response to this reviewer’s comment we have compared male and female (n=7) mice in the RAM task and found that their performance over the nine days of the task were very similar (Supplementary Fig. S1M). We also counted OL lineage cells in good- versus poor-performing females. Regrettably, there were only 2 good- and 2 poor-performers among the 7 females, the other 3 being intermediate; nevertheless, we found that the good-performing females, like good-performing males, had higher OLP proliferation, elevated OLP population and increased OL generation compared to poor-performers (Supplementary Fig. S1N, O, P). In future we will certainly use both males and females together, which will have the added benefit of increasing “n”.

Minor points:

Missing citations on page 12 line 6

We have fixed this (references 10-13).

In the last sentence of the summary, “...indicating a key role for adaptive OL genesis and myelination in cognitive processing” the conclusion is too vague. The authors do not show enough evidence that new myelination is occurring and implying this process is necessary for all cognitive processing is not an accurate conclusion from their data.

We changed the final sentence of the Abstract to:

“Remarkably, there was a strong positive correlation between working memory performance of individual mice and the scale of OLP proliferation and OL generation

during training, but not with the number or intensity of c-Fos⁺ neurons in the mPFC, underscoring the key role of OL lineage cells in cognitive performance.”

On page 2 paragraph 3, the authors state that there is preferential myelination of electrically active axons. However, the implication is that “inactive” axons are generally unmyelinated which is not necessarily true given the probability of an oligodendrocyte myelinating an axon is contingent on a variety of criteria. The study in reference was also achieved through pharmacological manipulation which may not be consistent with how myelination occurs naturally in-vivo.

These points made by the reviewer are undoubtedly true but we do not think that what we have written is controversial. We do not imply that all inactive axons are unmyelinated – we are only talking about adaptive (adult) myelination where it is a working hypothesis in many labs that axonal activity is communicated to OLPs and influences their proliferation/ differentiation/ myelination. This is fine-tuning the pattern of myelination that is mostly laid down during early postnatal development, possibly in an activity-independent manner. To make this more clear we modified the first sentence of the paragraph in question to say “**During adulthood,** OLPs and newly-forming OLs can detect and respond to electrical activity in the axons that they contact.”

On page 11 line 16, the authors state that the maze training results in EdU+/CC1+ OLs forming new myelin sheaths but do not quantifiably show this.

Our new experiments with *Tau-mGFP* (new Fig. 6) now make it much more convincing that additional new myelin is formed during maze training, so I hope that the reviewer will now accept this statement.

We hope that the extensive revisions described above will satisfy the reviewers so that our article can now be accepted for publication in Nature Communications.

Yours sincerely,

For all authors.

REVIEWERS' COMMENTS:

Reviewer #1 (Remarks to the Author):

The authors have addressed many, though not all, of the issues I raised with the previous version of the manuscript. The revised manuscript makes it clearer that working memory requires newly generated OL and subsequent myelination, and less (if any) neuronal activity per se. The assessment of neural circuit activity could have been more convincing using electrophysiological measures rather than c-Fos immunostaining. I believe that more could have been done in this regard, but this does not preclude the publication of this otherwise convincing study.

Reviewer #2 (Remarks to the Author):

All our comments are now satisfactorily addressed.

Reviewer #3 (Remarks to the Author):

While the authors did respond to all my comments/criticisms well and integrated the feedback into the manuscript, I do agree with Reviewer 1 that the findings are somewhat expected given their previous work (McKenzie et al., 2014) and that it would be more attractive if the mechanism driving these changes was described. The c-fos experiment does begin to try to answer the question about the relationship between neuronal activity and changes in myelination, however, this also doesn't provide substantially new information that hasn't been described in previous RAM studies (Milczarek et al., 2018, Van et al., 2000, Floresco et al., 1997). Overall, the work would be significantly more impactful if the authors would be able to describe a mechanism by which they think drives changes in myelination, and/or identify cell-type specific changes in myelination between homecage and experimental mice.

Shimizu, Nayar et al. NCOMMS-23-02481A. response to reviewers

Reviewer #1 (Remarks to the Author):

The authors have addressed many, though not all, of the issues I raised with the previous version of the manuscript. The revised manuscript makes it clearer that working memory requires newly generated OL and subsequent myelination, and less (if any) neuronal activity per se. The assessment of neural circuit activity could have been more convincing using electrophysiological measures rather than c-Fos immunostaining. I believe that more could have been done in this regard, but this does not preclude the publication of this otherwise convincing study.

We thank the reviewer for his/her positive response to our revisions. We accept that electrophysiology experiments would be interesting to complement the c-Fos immunolabelling but we strongly believe that this would take our study in a new direction outside of its current scope. This is something we want and plan to do in future.

Reviewer #2 (Remarks to the Author):

All our comments are now satisfactorily addressed.

Reviewer #3 (Remarks to the Author):

While the authors did respond to all my comments/criticisms well and integrated the feedback into the manuscript, I do agree with Reviewer 1 that the findings are somewhat expected given their previous work (McKenzie et al., 2014) and that it would be more attractive if the mechanism driving these changes was described. The c-fos experiment does begin to try to answer the question about the relationship between neuronal activity and changes in myelination, however, this also doesn't provide substantially new information that hasn't been described in previous RAM studies (Milczarek et al., 2018, Van et al., 2000, Floresco et al., 1997). Overall, the work would be significantly more impactful if the authors would be able to describe a mechanism by which they think drives changes in myelination, and/or identify cell-type specific changes in myelination between homecage and experimental mice.

We are glad that this reviewer accepts that we responded to all of his/her original comments/ criticisms/ feedback. As we said to Reviewer 1 in our previous rebuttal letter, we cannot agree that a role for OL generation specifically in working memory is "somewhat expected". We certainly do not think the striking proportionality we have demonstrated between working memory performance and OL dynamics could have been expected. As for the new c-Fos experiments that we have included in the present version of our paper, it was not our primary intention to advance understanding of, or provide new information about, neuronal activity patterns during RAM tasks, but rather to start to address Reviewer 1's specific comment about whether the increased OL lineage dynamics we observe during RAM training is simply a reflection of increased neuronal activity. The answer seems to be that it is not so simple, as the number of c-Fos+ neurons in the mPFC apparently does not change or even *decreases* as the number of OL lineage cells *increases*. We do agree that it would be amazing if we could deduce the mechanism through which OL lineage dynamics is increased during learning but we think this is unrealistic at the present time, at the relatively early stage of development of this field in general and with the specific expertise and technologies available in our lab. We will strive to understand mechanisms in future studies, but this will require us to develop new tools and approaches.